# Artificial transneurons emulate neuronal activity in different areas of brain cortex

Rivu Midya[1,8,9], Ambarish S. Pawar[2,9], Debi P. Pattnaik [3,9], Eric Mooshagian [4,9], Pavel Borisov [3], Thomas D. Albright [2], Lawrence H. Snyder [5], R. Stanley Williams [6], J. Joshua Yang [1,7] ✉, Alexander G. Balanov [3] ✉, Sergei Gepshtein [2] ✉ & Sergey E. Savel'ev [3] ✉

Rapid development of memristive elements emulating biological neurons creates new opportunities for brain-like computation at low energy consumption. A first step toward mimicking complex neural computations is the analysis of single neurons and their characteristics. Here we measure and model spiking activity in artificial neurons built using diffusive memristors. We compare activity of these artificial neurons with the spiking activity of biological neurons measured in sensory, pre-motor, and motor cortical areas of the monkey (male) brain. We find that artificial neurons can operate in diverse self-sustained and noise-induced spiking regimes that correspond to the activity of different types of cortical neurons with distinct functions. We demonstrate that artificial neurons can function as trans-functional devices (transneurons) that reconfigure their behaviour to attain instantaneous computational needs, each capable of emulating several biological neurons.

It is unlikely that hardware emulation of the whole biological brain can be realised in the near future. But emulation of reduced models of the brain appears to be within reach[1–3], offering much-needed insight into information processing in biological neural systems and opening new avenues for artificial intelligence. In accord with the neuron doctrine in neuroscience[4], the first step toward understanding complex neural computations is the analysis of single neurons and their characteristics, such as receptive fields and response fields.

The concepts of receptive field and response field have played a key role in neurophysiology[5], helping to understand how the brain processes information (Fig. 1). To the physicist, this concept is reminiscent of the concept of mean field, where interactions among many particles in an ensemble are approximated by an effective field acting on a pseudo-particle, representing the collective behaviour of real particles. Similarly, the interactions among neurons in a network are represented by the receptive/response field of a single neuron, enabling prediction of the collective activity of neural populations[5,6]. Here we use artificial memristive neurons to emulate activity of biological neurons in functionally different parts of the brain.

Memristors are among the most promising technologies for emulating[7–9] brain activity, which are used to develop artificial neurons and synapses in hardware[10]. These electronic circuit elements have a resistance that depends on the history of electrical stimulation. Defined axiomatically by Chua[7] and implemented in a physical device by Williams et al.[8,9], the invention of memristors spurred the development of a broad family of classical and quantum elements[11,12]. The state of these elements depends upon the history of current and voltage in electric circuits, paving the way for future reconfigurable electronics capable of surpassing Moore's scaling limits, emulating the

[1]Department of Electrical and Computer Engineering, University of Massachusetts, Amherst, MA, USA. [2]Systems Neurobiology Laboratories, Salk Institute for Biological Studies, La Jolla, CA, USA. [3]Department of Physics, Loughborough University, Loughborough, UK. [4]Department of Cognitive Science, University of California San Diego, La Jolla, CA, USA. [5]Department of Neuroscience, Washington University School of Medicine, St. Louis, MO, USA. [6]Department of Electrical and Computer Engineering, Texas A&M University, College Station, TX, USA. [7]Department of Electrical and Computer Engineering, University of Southern California, Los Angeles, CA, USA. [8]Present address: Department of Electrical and Computer Engineering, Texas A&M University, College Station, TX, USA. [9]These authors contributed equally: Rivu Midya, Ambarish S. Pawar, Debi P. Pattnaik, Eric Mooshagian. ✉e-mail: jjoshuay@usc.edu; a.balanov@lboro.ac.uk; sergei@salk.edu; s.saveliev@lboro.ac.uk

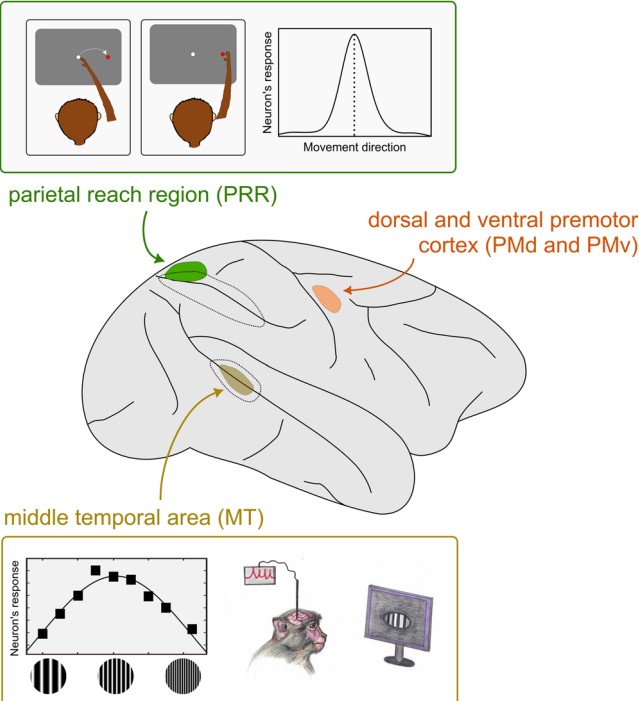

**Fig. 1 | Neuronal selectivity in monkey brain cortex.** A sketch of the monkey brain, in which three colours mark middle temporal (MT) area, parietal reach region (PRR), and premotor (PM) cortical area. (The same three colours are used in Figs. 2 and 5 to represent three populations of cortical neurons for comparison with the artificial neuron.) These cortical areas are responsible for different functions: visual perception in area MT (yellow), planning of movement in area PRR (green), and preparation of movement in area PM (orange). Neuronal selectivity in areas MT and PRR is illustrated in the insets. In both cases, the response of a single neuron (measured in the cortex of an alert animal) depends on the relations between properties of the neuron and properties of the stimulus (in sensory neurons) or the planned action (in pre-motor and motor neurons). Thus, in the bottom inset, the neuron is most excited when the animal views a luminance grating presented in the neuron's receptive field at an intermediate spatial frequency, and it is less responsive to high and low spatial frequencies. In the top inset, the neuron is most excited when the animal plans movement in a certain direction within the neuron's response field. The top panel of Fig. 1 is adapted from Mooshagian, E., Holmes, C.D. & Snyder, L.H. Local field potentials in the parietal reach region reveal mechanisms of bimanual coordination. Nat. Commun. 12, 2514 (2021). https://doi.org/10.1038/s41467-021-22701-3 under a CC BY license: https://creativecommons.org/licenses/by/4.0/. The left part of the bottom panel of Fig. 1 (the graph) is modified from Gepshtein, S., Pawar, A.S., Kwon, S., Savel'ev, S., Albright, T.D. Spatially distributed computation in cortical circuits. Science Advances 8, eabl5865 (2022). https://doi.org/10.1126/sciadv.abl5865, AAAS under a CC BY license: https://creativecommons.org/licenses/by/4.0/. The illustration of the monkey observing the monitor in the bottom panel of Fig. 1 was created by Irina Savelieva.

human brain, and enabling powerful analogue CMOS-integrated computing hardware[13,14].

Among the various prior realisations of artificial neurons, whether using memristive[15–17] or non-memristive elements[18–20], the overwhelming majority are designed to respond to electric stimulation by generating deterministic spiking (e.g.[17,21,22]). In contrast, biological neural systems are intrinsically stochastic due to random chemical and electrical fluctuations, as well as noisy inputs from other neurons originating in their complex dynamics within the network. If artificial neuromorphic hardware fails to accurately emulate the stochastic behaviour of biological neurons, it may struggle to achieve brain-level performance in solving diverse optimisation problems[23]. Additionally, artificial neurons may be unable to replicate certain brain functions[24–26] that critically depend on stochastic dynamics. While it is well known

that the complex dynamics of memristive neurons can produce a variety of spiking regimes[21,22], it remains unclear whether memristive neurons can mimic the stochastic dynamics of various types of neurons in the living brain.

In the following, we investigate the behaviour of inherently stochastic artificial neurons built using diffusive memristors and compare it with spiking activity of neurons in different regions of the brain cortex in awake animals (Fig. 1). These cortical regions participate in various cognitive functions, including the analysis of visual stimulation and the planning and control of hand movement. Despite the functional differences, the operations and information processing conducted by neurons in these different cortical regions share significant similarities. For instance, these neurons exhibit selectivity, i.e., they boost responses to certain features of stimulation or types of movement (Fig. 1 insets).

To compare the properties of biological neurons with their artificial counterparts, we use artificial neurons based on diffusive memristors. In these devices, clusters of Ag atoms ('Ag clusters,' Fig. 2B, C) diffuse between electrodes to form conducting filaments. The formation and rupture of these filaments are responsible for switching of the neuron's conductance[27–29]. We observe distinct spiking patterns in measurements of the artificial neuron and perform stochastic simulations of artificial diffusive neurons to reveal their distinct dynamical regimes. These regimes are associated with self-sustained or noise-induced spiking that drives the neurons' selective responses to stimulation. We then confirm all the predicted dynamical regimes in experiments with fabricated artificial neurons and compare the activity of these artificial neurons with the activity of biological neurons measured in three distinct cortical areas in awake macaque monkeys (Fig. 1). We find that a single artificial neuron can emulate the stochastic spiking behaviour of biological neurons in each studied cortical area by tuning the applied voltage, bath temperature and/or parameters of the neuromorphic circuit responsible for the characteristic time of artificial neuron (Fig. 2A).

We also demonstrate that artificial neurons exhibit more intense spiking response at their 'natural' frequencies, revealing frequency selectivity regulated by stimulation intensity, closely analogous to the biological selectivity observed in neurons of the visual cortex[6]. Additionally, we show how a single transneuron can process two signals simultaneously and perform phase detection by leveraging the dynamical multistability of its spiking regimes. This is evidence that a single artificial neuron can perform tasks that would typically require several biological neurons. We introduce the term 'transneuron' to indicate that the same physical device—an artificial neuron—can transition between the spiking characteristics of different types of biological neurons (Fig. 2A).

## Results
### Noise-induced and self-sustained spiking in artificial diffusive neurons
We fabricate diffusive memristors and use them to build transneurons (Fig. 2B, C), in which we measure their responses to input voltages (see 'Methods'). When a DC voltage above a certain threshold is applied, the artificial neuron generates spikes of current, which are similar to the spiking activity of biological neurons. To understand the spiking mechanisms of an artificial neuron, we simulate transneuron spiking dynamics using stochastic differential equations [Eqs. (1a–c); and Section 1 in Supplementary Information (SI)].

We investigate different spiking modes in artificial neurons by analysing how spiking depends on circuit parameters as well as external voltages and temperatures (Fig. 3). As we change the DC voltage, we observe dynamical patterns of current spikes that manifest different degrees of stochasticity. The experimentally observed spiking patterns range from quite regular to noisy, or to repetitive spike trains, as shown in the different columns of Fig. 3. These patterns

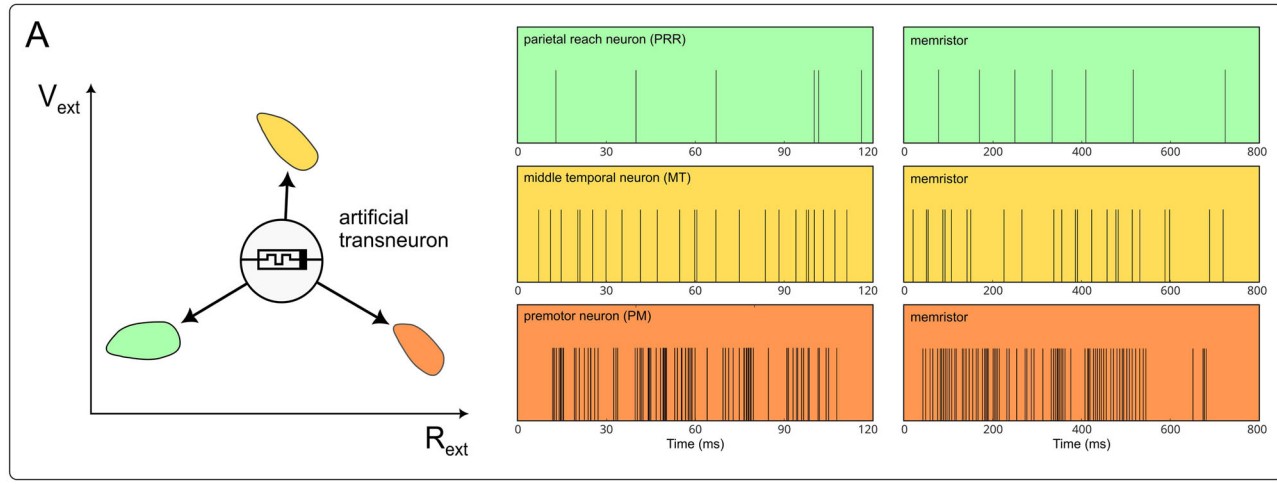

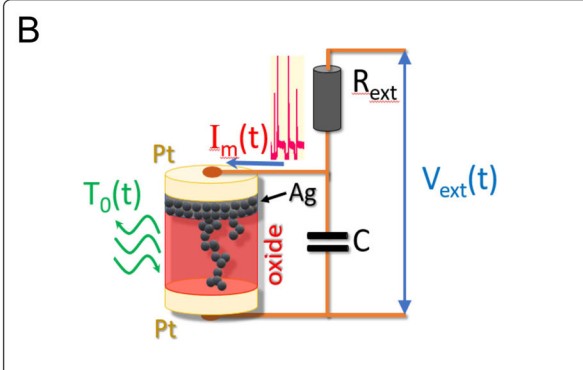

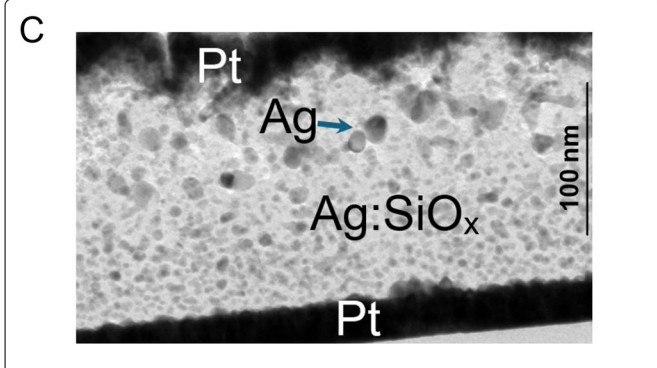

**Fig. 2 | Artificial transneuron. A** A conceptual representation of the transneuron moving in the parameter space defined by external voltage $V_{ext}$ and load resistance $R_{ext}$, reaching distinct regimes of stochasticity. In these regimes, the transneuron emulates the spiking behaviour of biological neurons in the cortical areas shown in Fig. 1, with matching colours. The three pairs of measured spikes at right illustrate distinct neural response patterns: irregular spiking in middle temporal (MT) area, more regular in parietal reach region (PRR), and bursting in premotor (PM) area. In each pair of plots of the same colour, the left plot represents the measured activity of a biological neuron, while the right plot represents the measured activity of the artificial transneuron in a distinct activity regime. **B** A sketch of the artificial transneuron, with the diffusive memristor represented by the cylindrical element at left. Ag clusters (black dots) diffuse between two Pt electrodes in an $SiO_x$ matrix (reddish medium), forming a filament that connects the memristor terminals.

When the memristor, with an internal or external capacitance $C$, is connected in series with an external resistance $R_{ext}$ and loaded by the external voltage $V_{ext}$, the system generates electric current spikes, $I_m$ (schematically shown in red as $I_m(t)$). The system can be dynamically controlled over time ($t$) by varying the external voltage $V_{ext}$ and the bath temperature $T_0$ of the $SiO_x$ matrix. This elementary circuit is referred to as an 'artificial transneuron' to reflect its ability to emulate the behaviour of biological neurons in different brain areas. **C** Cross-section TEM image of a diffusive memristor. Two platinum electrodes (Pt) can be seen as capacitor plates with the intervening $SiO_x$ matrix having droplet-like Ag clusters in it. Additional resistance can be either attached to Pt electrodes or realised by depositing a resistive material directly on the Pt electrodes. Twenty micrographs taken from different areas of the sample consistently display similar structural characteristics of Ag cluster distributions.

typically occur within specific intervals of input voltages, beginning at a certain threshold voltage, intensifying as the voltage increases, and then diminishing or disappearing at higher voltages (Fig. 3A–E or F–J). Changes in temperature also influence spiking, making the patterns noisier and/or more intense. Adjusting either the input voltage or the load resistance can lead to qualitative changes in the spiking pattern, such as transitions from regular spiking to spike trains.

For example, first consider measurement of transneurons stimulated by long pulses of a constant (DC) voltage (Figs. 3 and 4). At low DC bias voltage, no spiking is observed. As the voltage increases, the system first generates rare and random spikes, and then it evolves toward a regular spiking mode (Fig. 3A–C). A comparison of the measurement results and stochastic simulations is shown in Fig. 4A, D, respectively. This quite regular spiking regime persists in simulations even when the random noise in the system is completely suppressed. Indeed, our bifurcation analysis shows (see SI Section 3 and Fig. S1) that regular spiking occurs in this regime because the system develops a limit cycle via an Andronov-Hopf bifurcation at a certain threshold

voltage $V = V_{th1}$. A gradual increase in spiking intensity, as observed in our stochastic simulations and experiments (Figs. 3 and 4), is consistent with this dynamical mechanism.

As voltage increases further, spiking becomes sparse and irregular again (see results of measurements in Figs. 3D, E and 4B and stochastic simulations in Fig. 4E), consistent with the dynamical mechanism of disappearance of the limit cycle at the second Andronov-Hopf bifurcation at $V = V_{th2}$. Intriguingly, more intense and fairly regular spiking reappears at even higher voltages, shown in Fig. 4C for measurements and in Fig. 4F for stochastic simulations. In contrast to the regular spiking regime at lower voltages, this spiking regime does not arise if the contribution of noise in Eqs. (1a–c) is ignored. This fact suggests that heat can generate pseudo-deterministic forces due to thermophoresis and/or Seebeck effects[30,31], which produce another limit cycle.

A simple deterministic model [Eq. (2a–d)] accounts for the temperature-gradient forces and successfully predicts results of both stochastic simulations and experimental measurements of the appearance of the second spiking regime. The bifurcation analysis of

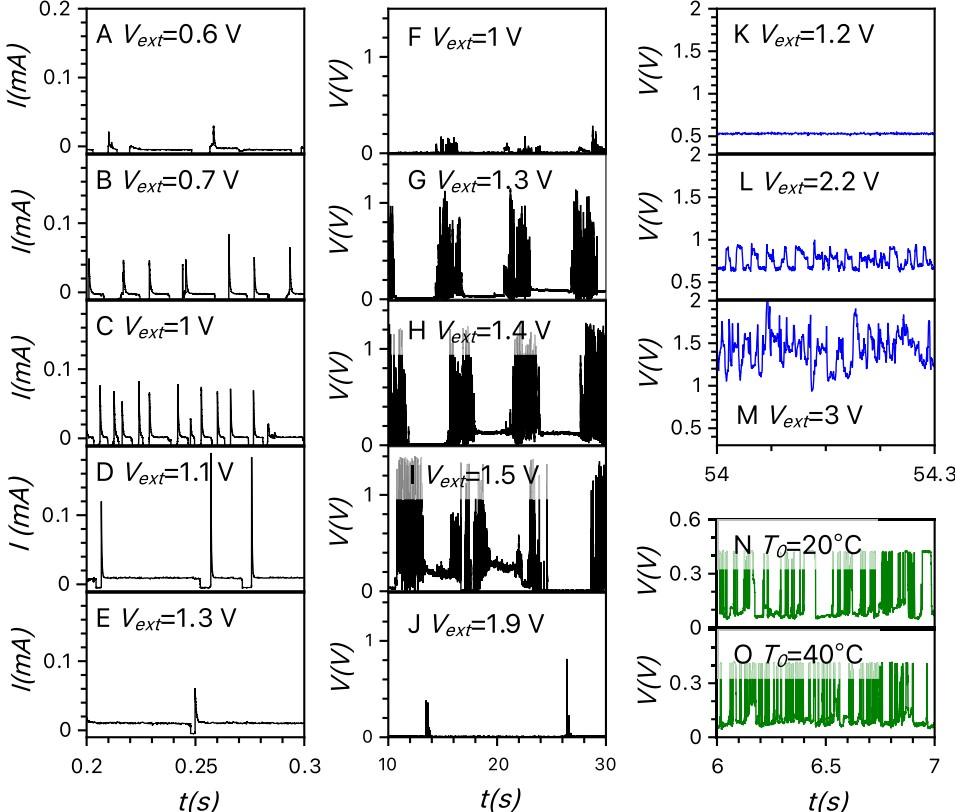

**Fig. 3 | Measured spiking activities of transneurons in different regimes under varying external voltage and temperature.** Left column **A–E:** Measured response of the transneuron in a PRR-like regime; the plots present measurements of current, $I$, in milliamperes (mA) versus time in seconds (s) at applied voltages, $V$, measured in volts (V) from 0.6 to 1.3 V. Within this voltage interval, we observe evolution of regular isolated spikes. Spiking appears at the voltages above 0.6 V, grows more intensive for 0.7 V and 1 V, then decreases for 1.1 V and stops for 1.3 V. (In these measurements, the external resistance was 68 kΩ and the capacitance was 10 nF. The memristor was fabricated using Method 1; see 'Methods'.) Middle column **F–J:** Bursting behaviour of voltage spikes is shown at intermediate voltages (1–1.9 V). Bursting appears at the voltage of about 1 V, develops with the increasing voltage

(1.3 and 1.4 V), followed by depletion of spiking in bursts at 1.5 V, and disappears at about 1.9 V. (In this transneuron, the external resistance was $R_{ext}$ = 65 KΩ and the external capacitance was C = 50 nF. The sample was fabricated using Method 2.) Right column (top three panels, **K–M**): MT-like spiking of a transneuron ($R_{ext}$=65Kohm and C = 50 nF, fabricated using Method 2). Here, spiking starts at a relatively high voltage threshold of about 2.2 V. Then spiking frequency grows as external voltage increases to 2.2 V and 3 V. The bottom two panels (**N, O**) at right show the influence of temperature on spiking (for temperatures, $T_0$, measured in degrees of Celsius, °C, from 20 to 40 °C), resulting in a further rise of spiking intensity in this transneuron (with $R_{ext}$ = 60 kΩ, C = 20 nF, fabricated by Method 2).

these deterministic equations reveals an additional dynamical spiking mechanism, originating from a distinct limit cycle that emerges through a nonlocal bifurcation at higher voltages, complementing the regular (periodic) oscillations observed at the lower voltages. The physical mechanisms underlying these two distinctive spiking regimes observed at low and high voltages differ: low-voltage oscillations result from the charging and discharging dynamics of circuit capacitance, whereas high-voltage oscillations stem from the slower heating and cooling cycle[32].

To evaluate the degree of stochasticity and regularity in these spiking regimes, we compare the power spectral density of the electrical current spikes observed in our simulations and measurements at three different voltages (measurements shown in Fig. 4G and simulations in Fig. 4H). At low voltages, the current spectral density exhibits clear maxima at specific frequencies, indicating the characteristic oscillation period (and its harmonics) of the artificial neuron. At intermediate voltages, the spectral maximum is suppressed, reflecting an irregular response with significant contributions across a wide range of frequencies. At high voltages, prominent but broad maxima reappear, indicating self-sustained yet noisy spiking.

Next, we vary the voltage slowly and linearly over time, in contrast to the previously described DC voltage pulses. This slow sweep of

voltage reveals that spiking patterns transition smoothly between regimes, indicating that the level of stochasticity can be precisely controlled within the same device by adjusting the bias voltage. These results are shown for measurements in Fig. 4I and for simulations with two identical Ag-clusters in Fig. 4J (see Section 3 in SI): spiking at low voltages, which shows quite regular pattern, is replaced by spike trains at higher voltages. Such spike trains are similar to the activity pattern commonly observed in biological neurons[33] (Section 5 in SI); their evolution with voltage is illustrated in Fig. 3F–J.

**Transneurons emulate visual, motor and pre-motor neurons in the monkey brain**

First, we compare the dynamics of artificial neurons with the spiking activity of biological neurons in two areas of the cerebral cortex—the middle temporal (MT) and parietal reach region (PRR)—in behaving rhesus monkeys. Neurons in these areas are known for their specialisation: they exhibit selectivity for visual stimuli in the MT area of visual cortex[34,35] and control for directed arm movements in the functionally defined PRR[36,37] (Fig. 1). In both sensory (MT) and motoric (PRR) areas, we measured spiking activity of individual neurons while monkeys performed a behavioural task (see 'Methods' for details).

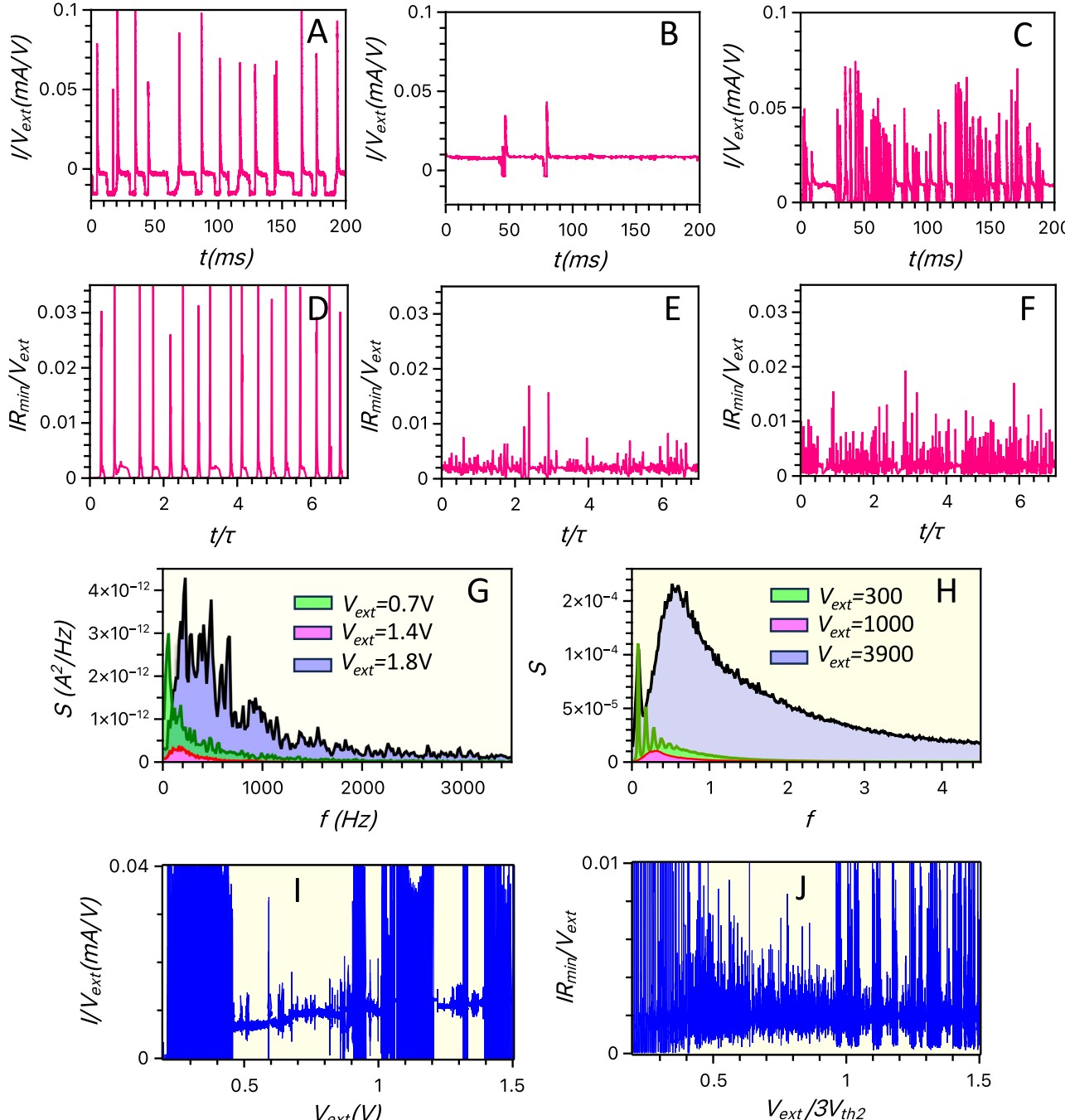

**Fig. 4 | Three spiking modes of the artificial diffusive neuron.** Experimentally measured spiking (**A**–**C**, pink curves) of electrical current, $I$ (in milliamperes, mA), in response to the external DC voltage $V_{ext}$, in volts (V), applied to the artificial neuron. Simulated spiking (**D**–**F**, pink curves) of current, normalised by $V_{ext}/R_{ext}$, with external resistance $R_{ext}$, using stochastic Eqs. (1a–c). Regular spiking is observed at low voltages in **A**, **D**, sparse irregular spiking is observed at intermediate voltages in **B**, **E**, and intermittent bursting is observed at higher voltages in **C**, **F**. The measured (**G**) and simulated (**H**) power spectra of system responses are represented by red, green, and black curves that correspond to spiking time series shown, respectively, in (**B**, **E**), (**A**, **D**), and (**C**, **F**). The plots reveal a clear spectral peak (green), indicating regular repetitive spiking in (**A**, **D**) at low voltages, a broad spectral maximum (black) for noisier bursting spiking in (**C**, **F**) at higher voltages, and a weak frequency dependence (red) of the spectra for sparse spiking in (**B**, **E**) at intermediate voltages, consistent with the spiking modes shown in (**A**–**F**). The measured (**I**) and simulated (**J**) stochastic spiking (blue traces) of current in the diffusive memristor changes gradually as the voltage $V_{ext}(t)$ varies slowly in time $t$ across the device. The bursting activity at high voltages is separated from regular spiking at low voltages by a regime of sparse spiking. These results demonstrate that voltage can be used to tune the activity of artificial transneurons. (Parameters of simulation are displayed in Supplementary Table 1 in SI).

We evaluate stochastic properties of spiking activity in these cortical areas using two metrics[38]. One is the coefficient of variation $CV_1$, which is the ratio of the standard deviation of inter-spike intervals (ISI) to the mean ISI. $CV_1$ is an index of stochasticity across the entire duration of the trial; it reveals the amount of periodicity in the system dynamics, and it quantifies the amount of irregularity generated by noise. For example, enhancement of stochasticity increases the standard deviation of ISI and thus yields a larger value of $CV_1$.

The second metric is the coefficient of variation $CV_2$, which is an index of local variability (or local time correlations) of spiking activity. We estimate $CV_2$ by first calculating the value of $2\left|\Delta t_{n+1} - \Delta t_n\right| / (\Delta t_{n+1} + \Delta t_n)$, where $n$ denotes the spike serial number and $\Delta t_n$ is the inter-spike interval separating $(n+1)th$ and $nth$ spikes; we then average the results over the duration of the trial. This metric ($0 \le CV_2 < 2$ by definition) is used widely for the analysis of spiking in different parts of the brain[38]; it describes correlations in sequential inter-spike intervals (ISIs) and it is useful for quantifying persistence of ISI changes. The lower the value of $CV_2$ the more persistent the correlations in the sequence of ISIs, allowing for more reliable readout of the encoded information. However, the lower $CV_2$ the less information can be stored in ISI sequences, suggesting that information processing is most efficient at intermediate values of $CV_2$. For the studied transneurons, our analysis shows that $CV_2$ can be tuned over a broad range between 0.1 and 1.9 (Fig. 5A), thus offering significant flexibility for task-specific computational needs. Nevertheless, to understand what values of $CV_2$ optimise task-specific information processing, one should consider specific computational protocols, larger neural networks, and various readout mechanisms (not studied here).

In Fig. 5A, we plot $CV_1$ and $CV_2$ metrics against one another for multiple MT (yellow squares) and PRR (green diamonds) neurons. The plot reveals that clusters of $(CV_1, CV_2)$ points that correspond to spiking activity of MT and PRR neurons occupy different regions in this parameter space. This finding reveals different stochasticity and sensitivity to stimulus excitation in these cortical areas. We find that these stochastic characteristics are not significantly affected by stimulus intensity indicating that they are mainly intrinsic to neurons (Fig. S5 in Section 7 of SI). In Fig. 5A, we also plot the $(CV_1, CV_2)$ points obtained from numerical simulations (grey hexagons) of our transneuron stochastic model and measurements from artificial neurons (small black circles) across various load resistance and DC voltage values. The hammer-like shape of the simulated $(CV_1, CV_2)$ cloud closely resembles the shape of the region occupied by the measured $(CV_1, CV_2)$ points of transneurons. The simulated and measured data for transneurons overlap significantly with the data representing the activity of biological neurons in areas MT and PRR. Specifically, the convex hull of $(CV_1, CV_2)$ points for transneurons measured at different external voltages, resistances, and capacitances, shown in Fig S6, covers about 70% of PRR $(CV_1, CV_2)$ points and 100% of MT $(CV_1, CV_2)$ points. Additionally, we simulated spiking activity to estimate $(CV_1, CV_2)$ characteristics under varying noise intensities (see Section 4 of SI). The results show that a modest increase in noise shifts the $(CV_1, CV_2)$ point distributions, causing only a minor reduction in the overlap between the $(CV_1, CV_2)$ distributions of biological neurons and transneurons.

The significant overlap of the measured $(CV_1, CV_2)$ points for the transneuron with the $(CV_1, CV_2)$ clouds for MT and PRR neurons is evidence that tuning the load resistance and DC voltage applied to the artificial neuron can produce distinct stochastic spiking activities that emulate the behaviour of these biological neurons. Such tunability arises from the competition between several dynamical attractors, each affected differently by noise and characterised by distinct basins of attraction. The multiple bifurcations observed in transneurons can be associated with the phenomenon of 'edge of chaos'[39], highlighting their remarkable capacity for dynamic reconfiguration.

The finding that transneurons can produce spiking with $CV_1$ values even higher than the typical range for MT neurons, albeit with lower $CV_2$, suggests another type of spiking behaviour characterised by a combination of high persistence in successive ISIs and high overall randomness. We expect such behaviour from bursting neurons[33]. To explore this hypothesis, we analyse transneuron dynamics across a broader parameter range than above, including the temperature dependence of spiking (see Fig. 3N, O for an example). As predicted, we find a distinct patterns of bursting activity (Fig. 3F–J). Notably, increasing the bath temperature results in noisier bursting, characterised by shorter and more irregular quiet intervals (see Fig. S2 in Section 5 of SI). Additionally, by varying the external voltage, we observe that bursting occurs only within a specific voltage range (Fig. 3F–J), allowing the bursting regime to be switched on or off, or even tuned to specific inter-train interval values.

We compare these results with the bursting activity of biological cortical neurons recorded in the monkey premotor cortex (PM) using publicly available spiking data[40]. Intensive spiking activity ('bursting') alternates with quiet intervals in both biological PM neurons (Figs. 2A and S3A), measured transneurons (e.g. Fig. 3F–J), and simulated transneurons (Figs. 5D and S2B, C) within a specific window of external voltages. We add the $(CV_1, CV_2)$ points for bursting PM neurons to Fig. 5A (orange triangle), revealing significant overlap with the simulated and measured $(CV_1, CV_2)$ points for artificial transneurons. Indeed, the convex hull of the measured $(CV_1, CV_2)$ points for transneurons covers 100% of PM $(CV_1, CV_2)$ points (see Fig. S6 in SI). This finding suggests that transneurons can also emulate features of spiking activity in the PM cortex.

Next, we demonstrate that by setting specific values for the load resistance and capacitance of the transneuron, and varying the external voltage within a certain range, it is possible to target a preselected region of the $(CV_1, CV_2)$ stochastic characteristic. For example, the measured $(CV_1, CV_2)$ points for a transneuron with a load resistance of 65 kOhm and an external capacitance of 1 nF, when the external voltage varies between 0.6 and 0.8 V, fall within the $(CV_1, CV_2)$ region corresponding to the stochastic behaviour of MT neurons (Fig. 5A1). This demonstrates that, within this external voltage range, the transneuron accurately mimics the spiking behaviour of MT neurons, with its stochastic features remaining stable despite modest voltage variations. The distribution of yellow and black points suggests that these voltage variations enable the transneuron to exhibit stochastic activity encompassing the entire $(CV_1, CV_2)$ cloud.

To complete the comparison of stochasticity between the transneuron and biological neurons, Fig. 5A2 shows that a transneuron with a load resistance of 70 K$\Omega$ and an external capacitance of 100 nF exhibits a gradual drift in the measured $(CV_1, CV_2)$ points between distinct $(CV_1, CV_2)$ regions corresponding to different types of biological neurons. These transneuron points shift from the region corresponding to PRR neurons to the region corresponding to PM neurons as the voltage changes from 1.06 to 1.22 V. Specifically, half of the measured transneuron $(CV_1, CV_2)$ points for the same neuron fall within the PRR cluster, while the other half fall within the PM cluster. This highlights the transneuron's ability to emulate biological neurons from different cortical areas by simply varying the external voltage applied to the same fabricated device.

Maps of stochastic characteristics, such as the map of $CV_1$ and $CV_2$, are instrumental for identifying circuit parameters and DC voltage settings that enable artificial neurons to emulate the stochastic characteristics of specific types of biological neurons, producing the desired response. For instance, Fig. 5B shows the simulated $CV_1$ as a function of load resistance ($R_{ext}$) and external voltage ($V_{ext}$). The artificial neurons replicate the spiking statistics of PRR neurons at lower $V_{ext}$ and $R_{ext}$ values, corresponding to the first spiking regime (Fig. 4A, D). At higher voltages, where noisy bursting occurs in the second spiking regime (Fig. 4C, F), the $CV_1$ values of artificial neurons align with those of PM neurons (see timestamps in Fig. S3A in SI). MT-like behaviour in artificial transneurons is observed at both higher voltages and higher resistances.

Finally, to visualise spiking patterns in different regions of the $(CV_1, CV_2)$ space, we present examples of simulated spiking conductance in Fig. 5C–E. These examples correspond to specific $(CV_1, CV_2)$ points selected from Fig. 5A, whose stochastic characteristics represent PRR, PM and MT neurons.

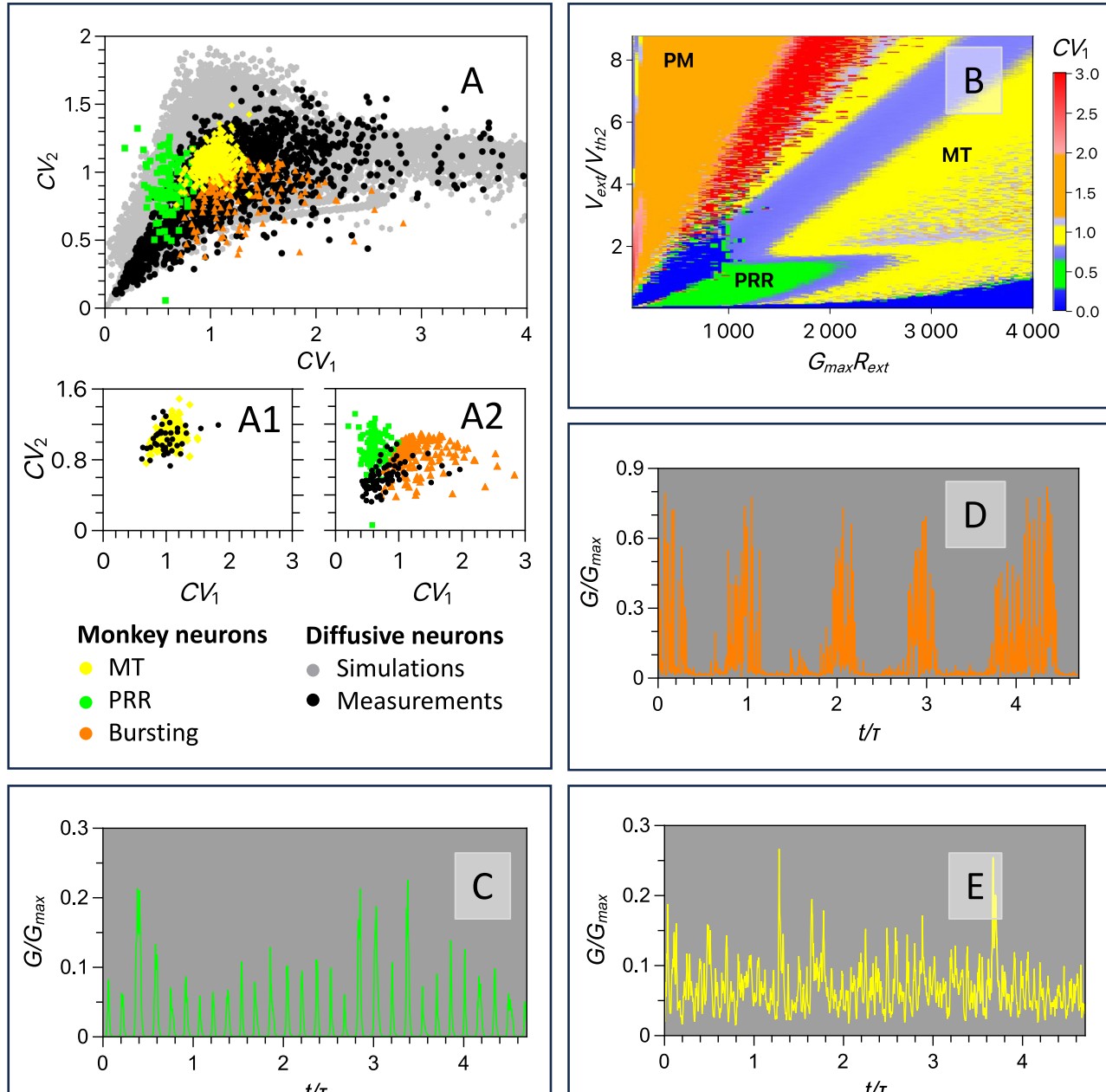

**Fig. 5 | Comparison of biological and artificial neurons. A** Scatter plots of the coefficients of variation CV$_1$ and CV$_2$ obtained (i) by simulating the one Ag-cluster model in Eqs. (1a–c) (grey hexagons), (ii) by measuring of transneurons (black circles), and (iii) by spiking analysis of biological neurons sampled from cortical middle temporal area (MT, yellow diamonds), parietal reach region (PRR, green squares), and premotor area (PM, orange triangles) in macaque monkeys. Panel A1 shows an overlap between the (CV$_1$, CV$_2$) points measured in both MT neurons and the transneuron for $R_{ext} = 65\,k\Omega$ and $C = 1\,nF$ within a voltage interval of 0.6−0.8 V. This plot shows that nearly all transneuron measurements occur within the MT cloud and are well distributed within it. Panel A2 shows that a voltage sweep from 1.06 to 1.22 V for a transneuron with $R_{ext} = 70\,k\Omega$ and $C = 100\,nF$ shifts the measured (CV$_1$, CV$_2$) points of the same transneuron between the PRR and PM clouds. This plot demonstrates the transneuron's ability to transition between the stochastic regimes inherent to biological neurons in different regions of the

monkey cortex. **B** The coefficient of variation CV$_1$ is presented as a colour map in a space of external resistance $R_{ext}$ (in units of $1/G_{max}$, with the maximum memristor conductance $G_{max}$) and external voltage $V_{ext}$ (in units $V_{th2}$ obtained at $G_{max}R_{ext} = 500$). The map was obtained by simulations of the artificial neuron [Eqs. (1a–c)] with varying load resistance and applied voltage. The regions where the CV$_1$ of the simulated transneuron corresponds to the CV$_1$ of biological neurons in cortical areas MT, PRR, and PM are shown respectively in yellow, green, and orange, matching the colours used to label these cortical areas in Figs. 1A and 5A. **C**–**E** Examples of simulated spiking of transneuron conductance for (CV$_1$, CV$_2$) = (0.64. 077) (green curves), (1.9, 0.94) (orange curves), and (1.01, 1.05) (yellow curves), which correspond to the (CV$_1$, CV$_2$) points emulating MT, PRR, and PM neurons respectively, displayed in (**A**). (Simulation parameters are listed in Supplementary Table 1).

## Transneuronal computations

Biological and digital information processing follows different computational principles. Digital processing is implemented by sequences of logic gates applied to digitised information. Biological processing

involves parallel spiking activity elicited by analogue stimulation[41], which is sequentially filtered and followed by decision-making. Instead of logic gates, neural computations are mediated by filtering mechanisms[42], where selectivity is reflected in the spiking activity that

is enhanced or suppressed depending on the stimulus content. This dynamics modifies synaptic connections between neurons, enabling the system to encode past experiences and forming the basis for learning[43,44]. While synaptic weight adjustment has been implemented in artificial neural networks, here we focus on two other aspects of neural computation: (a) information processing through selective response to stimulation and (b) transitions between coexisting dynamical regimes. Both these aspects can be realised within a single artificial transneuron.

## Computation by selectivity

We investigate the selectivity of artificial neurons by applying a voltage signal $V_{ext}(t)$ comprising both DC and AC components ($V_{ext} = V_{DC} + V_{AC}\cos\omega t$, shown as blue curves in Fig. 6A, B) and measuring as well as simulating the electrical current response (red curves in Fig. 6A, B). DC and AC components of the signal allow one to (i) mimic the neuron's rest state (with spontaneous or non-stimulus-induced spiking activities) via DC voltage input, $V_{DC}$, and (ii) supply stimulus information via the frequency content of the AC signal, $V_{AC}\cos\omega t$. We first analyse the neuron's spiking behaviour as a function of the stimulus frequency ($\omega$) at a fixed stimulus magnitude ($V_{AC}$), which corresponds to stimulus intensity. This experimental approach draws inspiration from a common method used to measure the selectivity of visual cortical neurons, employing windowed drifting luminance gratings as optical stimuli[35].

To track the evolution of ISI distributions (Fig. 6C, D), we accumulate spiking statistics over time. This approach is analogous to information gathering in biological diffusion-decision models[26]. The colour map in Fig. 6C, D depicts the probability density of ISIs as a function of the AC stimulus period ($t_p = 2\pi/\omega$), which is the inverse of stimulus frequency. The red, white, and blue colours represent the respective high, medium, and low probability densities, respectively. The dotted green curve marks the most probable ISI, while the yellow dashed line indicates the ISI value matching the stimulus period. Over the range of the $t_p$ values close to the reciprocal of the neuron's natural spiking frequency (140 Hz at 1 V for the fabricated transneuron with $V_{AC} = 0$), the peak of the ISI distribution aligns with the stimulus period. Within this range, the ISI distribution narrows and exhibits a higher peak compared to when $t_p$ is away from the inverse of the natural spiking frequency. This behaviour reflects the neuron's frequency selectivity (see red regions in the measured and simulated 2D histograms in Fig. 6C, D). Such a selective activation of the transneuron resembles the phenomenon of synchronisation[45] observed in deterministic oscillator-based computation[46]. In our case selective activation is realised in an essentially stochastic system.

To clarify the relationship between stochastic synchronisation in transneurons and the selectivity of biological neurons, we note that the selectivity of biological neurons is typically determined by identifying the stimulus feature that elicits the maximum firing rate in response to a particular feature of the stimulus (Fig. 1). In Fig. 6E, we plot the firing rate of an MT neuron against the time-oscillation period ($t_{MT}$) of a visual stimulus (drifting luminance grating) at various luminance contrasts. The spike rate peaks at a specific value of $t^*_{MT}$, representing the neuron's preferred temporal frequency[6,33]. Interestingly, the preferred time-period ($t^*_{MT}$) shifts toward lower values as the stimulus contrast increases, mirroring the previously observed shift in the preferred spatial frequency of these neurons[6,35].

To compare the selectivity of biological neurons with the selectivity of artificial neurons, we plot the spike rate of transneurons as a function of the AC-voltage time period (here representing the intensity of simulation, analogous to the luminance contrast of visual stimulation of MT neurons). Similar to biological neurons, the spike rate of transneurons reaches a maximum (Fig. 6F, G) at a specific voltage oscillation period, $t^*_p$. Notably, both experimental data (Fig. 6F) and simulations (Fig. 6G) show a sharpening of the ISI distributions

(Fig. 6C, D) for $t^*_p$, where the maximum spiking rate is observed. Longer spiking sequences recorded in simulations reveal a shift in the maximum spiking rate as the AC-voltage amplitude ($V_{AC}$) increases. Specifically, $t^*_p$ shifts to lower values with increasing $V_{AC}$, mimicking the shift of $t^*_{MT}$ in MT neurons with increasing stimulus contrast (Fig. 6E). This finding highlights even deeper similarities in the non-linear spiking regimes of transneurons and biological neurons, extending beyond the so-called classical concept of receptive field, despite significant structural and parametric differences. These include: (a) visual stimulation does not directly excite MT neurons but rather through several intermediary stages in the neural visual pathways, and (b) the temporal frequency of visual stimulation is significantly lower than the spiking frequency of MT neurons.

## Computation by signal comparison

Signal comparison, particularly phase comparison, is widely used in biological systems. Notable examples include the computation of binocular disparity, which supports stereoscopic vision[47], and the computation of the speed of moving visual stimuli[48]. Signal comparison also plays a key role in many higher-order information processing tasks[49,50]. Beyond biological systems, signal comparators are extensively used, e.g. in communication technologies[51] and interferometry[52].

Here we propose an innovative computational approach for comparing the relative phases of two signals, leveraging the artificial neuron's ability to dynamically change its function during information processing. Specifically, we demonstrate how reconfiguring the dynamical regimes of spiking activity enables efficient multi-signal analysis. To explore how input voltage governs the 'transitions' between distinct dynamical states in the one-Ag-cluster transneuron model, we present a phase diagram of the spiking regimes obtained for both stochastic [Eq. (1a–c)] and deterministic [Eq. (2a, d)] sets of equations in Fig. 7A. The figure illustrates the time evolution of Ag-particle position ($x$), which determines the memristor's conductance as the external voltage ($V_{ext}$) changes slowly. Simulation results for the stochastic and deterministic equations are shown as light grey and orange lines, respectively. The position ($x$) represents the location of the Ag nanoparticle cluster within the gap separating the tip of the nearly completed conducting filament from the memristor electrode (see Section 1 in SI). The trajectory of $x(t, V_{ext}(t))$ exhibits hysteresis behaviour (dynamical multistability) as the applied voltage $V_{ext}$ first increases and then decreases. Hysteresis occurs because bifurcation depends not only on the external conditions (e.g. voltage or temperature) but also on the 'system trajectory,' which is determined by dynamical variables such as Ag-cluster location or local voltage. For this reason, when the applied voltage increases from the state where the Ag-cluster is attached to the tip of the filament, the system follows a different sequence of bifurcations compared to when the applied voltage decreases, and the Ag-cluster returns from the equilibrium position near the memristor terminal to the pillar tip. This complex dynamics results in two distinct regions (labelled I and III in Fig. 7A), each corresponding to a characteristic self-sustained spiking regime. These regions are separated by regions of noise-induced spiking (labelled II and IV in Fig. 7A), where the particle fluctuates either near one edge of the gap or near an equilibrium point of the system.

From these simulations and the bifurcation analysis, we conclude that within a specific range of external voltages ($V_{ext}$), the spiking regime coexists with the non-spiking (noise-induced) regime of Ag-cluster fluctuations (e.g. the voltage interval corresponding to the overlap of regions I and IV, indicated by the blue arrow in Fig. 7A). This multistability allows information to be stored in the coexisting dynamical states. Information stored in such states can be processed by switching between these states, as illustrated by the hopping indicated by the dark red arrow in Fig. 7A. This switching between spiking and non-spiking dynamics is triggered by thermal noise, whose intensity

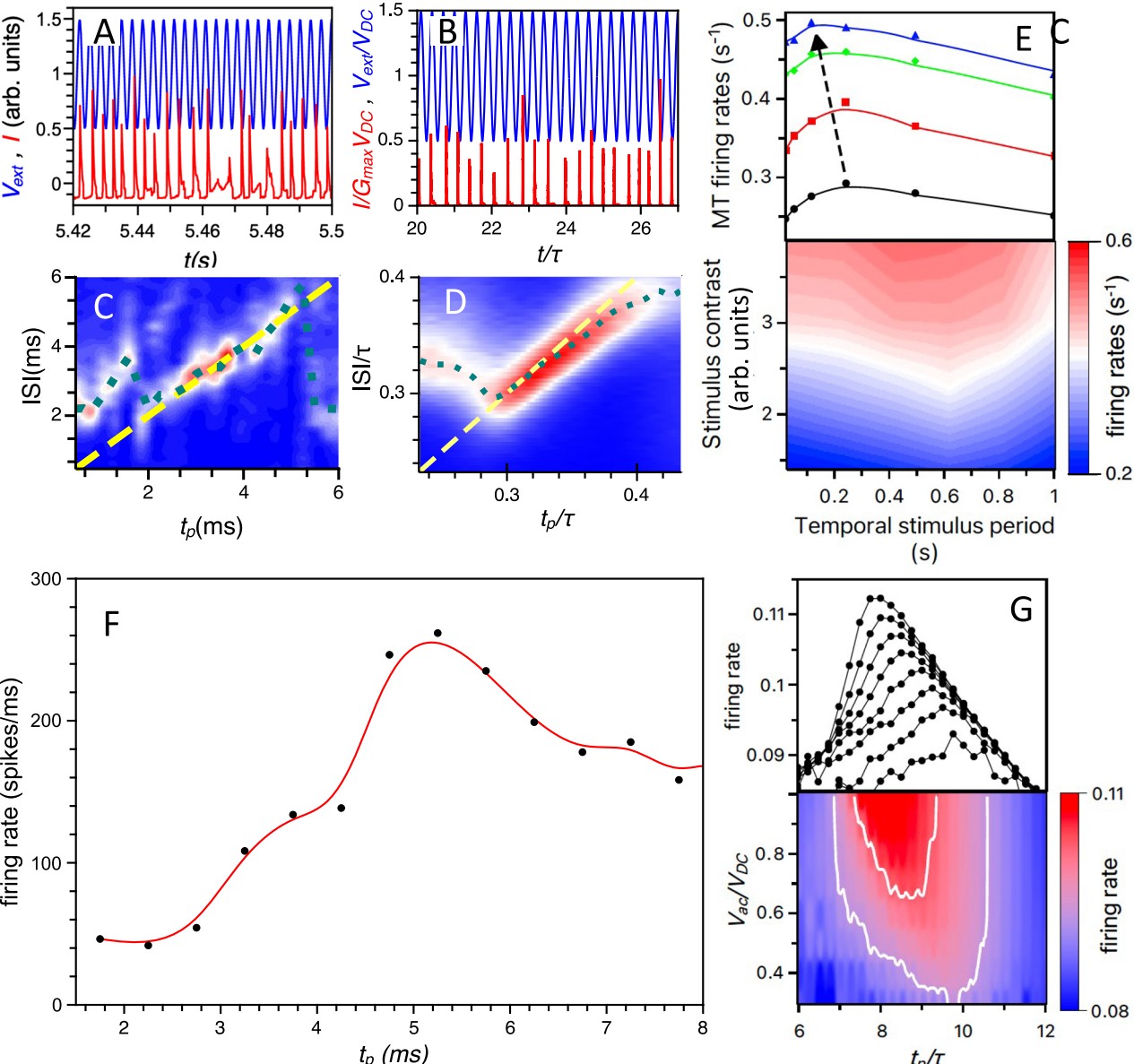

**Fig. 6 | Neuron selectivity. A** The input voltage signal, $V_{ext} = \left(1 + 0.5 \cos\left(2\pi t/t_p\right)\right)$, is shown in blue for $t_p = 7$ ms, with voltage in volts (V) and time in milliseconds (ms) [DC voltage $V_{DC} = 1\,$V, AC voltage $V_{AC} = 0.5\,V_{DC}$, external resistance $R_{ext} = 70\,$kOhm, external capacitance $C = 100\,$nF]. The measured current response is shown in red, in microamperes (μA). **B** Similar to (**A**), but showing the simulated current I(t) (in red) in response to external voltage $V_{ext}$ (in blue), with $V_{DC}$ equal to the applied voltage corresponding to the self-sustained, no-bursting spiking regime (shown in Fig. 2D) and for $t_p$ corresponding to the most probable ISI time. Here, the simulated time and ISI are displayed in units of $\tau = R_{ext}C$. Heat-Map contour plots of two-dimensional ISI histograms illustrating neuronal selectivity for the measured (**C**) and simulated (**D**) artificial diffusive neurons in the plane of stimulus period $t_p$ (oscillation period) and inter-spike interval (ISI) values. The yellow dashed lines represent the 'perfect selectivity conditions' (ISI=$t_p$) while the dotted green curves show the relationship between the most probable ISI and the stimulus period $t_p$. **C, D** can be interpreted as temporal receptive fields of artificial neurons. **E** The measured spiking rate of MT neurons as heat-map contour plot (bottom) or a function of the stimulus time-oscillation frequency at different luminance contrasts (from black-red-green-blue corresponding growing contrast) (top). The maximum rate shifts towards higher frequency (lower oscillation periods) as stimulus contrast increases (black arrow). **F** The measured spiking rate of the transneuron plotted against the AC-voltage period (black dots connected by B-spline line). The rate exhibits a clear maximum at a specific time ($t_p = t^*_p$) analogous to the spike rate maximum of MT neurons in (**E**). **G** The simulated transneuron spiking rate as a function of the AC-voltage period and amplitude. The bottom panel shows a heat-map contour plot, while the upper panel shows individual traces of the functional relationship r($t_p$) at different $V_{AC}$. The period $t^*_p$, corresponding to the maximum spiking rate, shifts to lower values, consistent with the shift of the maximum spiking rate of MT neurons shown in (**E**). (Simulation parameters are provided in Supplementary Table 1 in SI).

depends on the temperature. As noise increases, the likelihood of the Ag particle escaping its current dynamical state rises. To quantify this escape, we define the residual times the system spends either near the fixed point (non-spiking) or in the self-sustained limit cycle (intense spiking). These residual times determine the average spiking rate and are influenced by both temperature and external voltage.

To clarify the mechanism governing switching between different spiking regimes, we analyse how the basins of attraction for distinct dynamical states change with applied voltage when the system is in the metastable region. Using deterministic Eq. (2a–d), we simulate the system and find that the transneuron dynamics in the absence of noise are drawn either to the equilibrium point or the limit cycle, depending

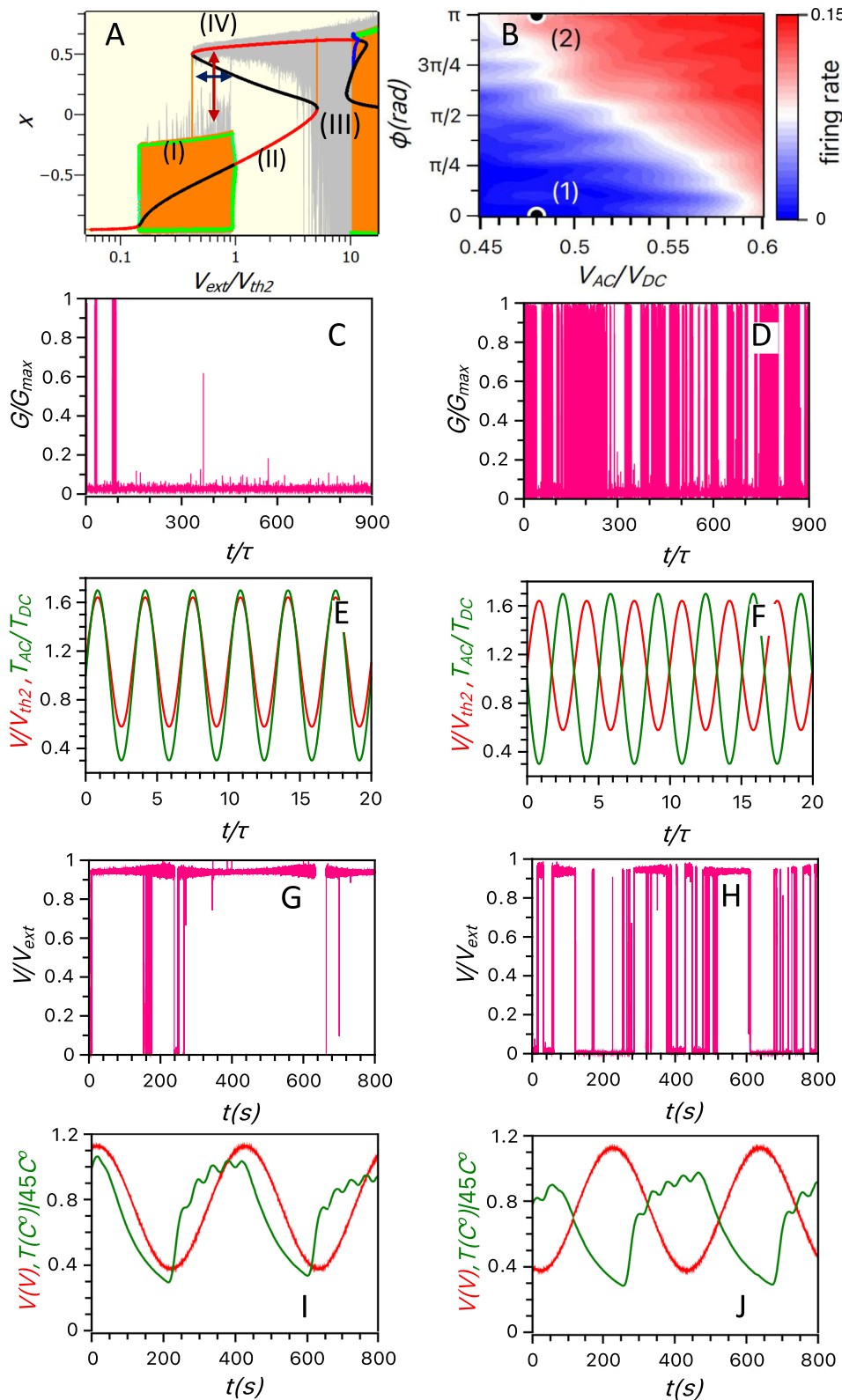

on the initial position of the Ag particle, $x(t=0)=x_0$ (see Fig. S1C in SI). The boundary separating regions of initial positions that lead to either spiking or non-spiking dynamics is formed by stable manifolds of the unstable equilibrium points. In the bifurcation diagram (Fig. 7A), this boundary is indicated by black dots $x_0(V_{ext})=x_c(V_{ext})$ (unstable fixed points), which separate the basins of attraction for the two stable solutions. The larger is the basin of attraction, the greater the noise

required for the system to switch to another state. The applied voltage controls the size of these basins (e.g. the basin of the equilibrium point disappears at the left tip of the dark blue horizontal arrow in Fig. 7A), while the temperature determines the intensity of the noise driving transitions between basins. For example, consider the moment when the applied voltage settles the transneuron in an equilibrium (non-spiking) state with a small basin of attraction. If the temperature is high

**Fig. 7 | Transneuron signal phase comparator. A** Phase diagram $(V_{ext}(t), x(t))$ for the one-Ag-cluster model of a diffusive neuron is obtained by slowly ramping the external voltage $V_{ext}$ and simulating trajectory $x(t)$ using stochastic Eq. 1 (grey curves) and deterministic Eq. 2 (orange curve). Both curves exhibit hysteresis. Two self-sustained spiking modes (regions I and III) occur at low and high voltages, separated by regions II and IV, corresponding to noise-induced sparse spiking. From the bifurcation analysis of Eq. 2, we obtained stable (red) and unstable (black) fixed points, where nearby trajectories converge or diverge exponentially. This analysis predicts two stable limit cycles, attracting nearby trajectories. The cluster position, $x(t)$, on the cycles oscillates between maxima and minima, represented by the green points. The bifurcation analysis of Eqs. 2 predicts an unstable limit cycle (with its maxima and minima shown by blue points), which repels nearby trajectories. For low-voltage spiking (regime I), there exists a range of voltages ($V_{ext}$, indicated by a blue arrow), where spiking coexists with fixed points (Region IV). **B** A heat-map contour plot of simulated average spiking rates generated by an artificial neuron (Eq. 1) excited by two low-frequency periodic signals: oscillating voltage [$V_{ext} = V_{DC} + V_{AC} \sin \omega t$] and oscillating bath temperature [$T_0 = T^*(1 + a_T \sin(\omega t + \phi))$]. The contour plot demonstrates a dependence of the spiking rate on the relative phase $\phi$ between these two signals. **C–F** Two examples of realisations of simulated spiking of conductance $G(t)$ normalised by its maximum value $G_{max}$ (corresponding to points 1 and 2 in (**B**)) are shown in pink in **C**, **D** for in-phase (**E**) and antiphase (**F**) signals, with $T_0(t)$ rendered in green and $V_{ext}(t)$ rendered in red. Minimal spiking is observed for the in-phase signal, and significant spiking for the antiphase signal. Measured spiking (pink curve) of voltage $V$ is shown in (**G**, **H**) for external voltage and temperature, varied in phase (**I**) and anti-phase (**J**); curves $T_0(t)$ and $V_{ext}(t)$ are shown in green and red, respectively. Consistent with simulations in (**C**, **D**), measured spiking is significantly suppressed for in-phase voltage-temperature oscillations (**G**) compared to the intense spiking for anti-phase oscillations (**H**). Simulation parameters are provided in Supplementary Table 1 in SI.

at the same moment, the probability of switching to another state is high, reducing the transneuron's residual time in that state. In contrast, if the temperature is low at this moment, the Ag-cluster is more likely to remain in this non-spiking state, leading to a longer residual time. However, when the voltage settles the transneuron in an equilibrium state with a large basin of attraction, temperature becomes much less relevant, as even high noise (temperature) is unlikely to switch the transneuron to another spiking mode. This interplay between the voltage-controlled size of the basin of attraction and the temperature-controlled noise intensity demonstrates how two signals, encoded in the $V_{ext}(t)$ and $T_0(t)$ time series, influence the transneuron's residence times in either intensive or rare spiking states. As these residence times determine the average spiking rate, this analysis establishes a direct connection between the two input time signals and the average spiking intensity.

The preceding analysis of bifurcation mechanisms underlying distinct nanocluster dynamics demonstrates how various dynamical states and the control of transitions between them could be harnessed to develop a neuromorphic two-signal phase comparator (a 'phase detector') using a single transneuron. Two input signals can be encoded in the external voltage $V_{ext}(t) = V_{DC} + V_{AC} \sin \omega t$ [the first time-dependent signal], and the bath temperature, $T_0(t) = T^*(1 + a_T \sin(\omega t + \phi))$ [the second signal]. The DC and AC components of these signals are $V_{DC}$ and $V_{AC}$ for the external voltage, and $T^*$ and $T^* a_T$ for bath temperature. Both input signals have the same frequency $\omega$, which is much lower than the average reciprocal ISI. The temperature signal is shifted by phase $\phi$. For $V_{DC} - V_{AC}$ within the region of multistability (indicated by the blue horizontal arrow in Fig. 7A), the spiking intensity (Fig. 7B) depends on the relative phase $\phi$, which is essential for phase detection. The spike rate (the computational output) reaches a minimum when the two signals are in phase ($\phi = 0$) and a maximum when the signals are in antiphase ($\phi = \pi$) (Fig. 7C–F). This behaviour reflects an interplay between noise-induced and self-sustained spiking. To qualitatively interpret this dependence, we need to examine the temperature at specific time points when the basins of attraction for (i) non-spiking and (ii) spiking states become very small, allowing noise to trigger an escape from these regimes. When the temperature and the external voltage oscillations are in antiphase, thermal noise promotes the system's transition into the self-sustained spiking regime [the low branch of $(xt, V_{ext})$ in the multistability region in Fig. 7A]. Indeed, for antiphase signals, the external voltage has the lowest value ($V_{ext} = V_{DC} - V_{AC}$) when the temperature is at its maximum [$T_0(t) = T^*(1 + a_T)$]. If the system at this instant resides in a non-spiking state (upper red branch in Fig. 7A), it is likely to switch to the spiking regime because the basin of attraction is small (due to low voltage) and the noise level is high (due to high temperature). Moreover, for $\phi = \pi$, when the voltage has its maximum value

($V_{ext} = V_{DC} + V_{AC}$), corresponding to the smallest basin of attraction in the spiking regime, the temperature is at its lowest. Therefore, the artificial neuron has a low chance of switching back to the non-spiking state because, even when the attraction region is small, the temperature is also low, reaching its lowest value $T_0(t) = T^*(1 - a_T)$, which produces insufficient noise for the transneuron to escape from the spiking state, switching back to non-spiking. This interplay of temperature-induced hopping between spiking regimes, on one hand, and the voltage-induced change in the basins of attraction for spiking states, on the other hand, causes the artificial neuron to remain in the spiking state significantly longer during antiphase ($\pi$-shift) oscillations of voltage and bath temperature. In contrast, for in-phase (0-shift) oscillations of external voltage and temperature, the same mechanism facilitates easier transitions from spiking to non-spiking regimes while suppressing reverse switching. Consequently, the transneuron spends much more time in the non-oscillating state when the phase shift is zero.

We conduct an experimental test to evaluate whether a single diffusive neuron can perform signal comparison. The neuron's spiking response is measured while varying the external voltage and background temperature in-phase (Fig. 7I) and anti-phase (Fig. 7J). The experimental results (Fig. 7G, H) closely match the simulations (Fig. 7C, D), demonstrating significantly more intense spiking during anti-phase oscillations of voltage and temperature compared to in-phase oscillations. For proof-of-concept, as shown here, the entire diffuse memristor was periodically heated. However, achieving energy efficiency would require localised heating, which could be implemented using several modern techniques. One of the most practical implementations of this concept could involve a nanoscale thin-film Joule heater[53]. Such systems could localise temperature oscillations to scales as small as a single memristor filament or even part of the filament. Since the heated region would be small (potentially involving just a few Ag clusters), the required power consumption and related energy loss could be very low. Moreover, the signal encoded in temperature modulations could, in principle, be implemented using a near-field enhancer of the laser field, creating hotspots as small as 10–100 nm[54,55]. However, this approach would increase the complexity of fabricating transneurons and neuromorphic devices as a whole. The energy loss per signal cycle can be estimated as $C_{Ag} \rho_{Ag} N (4\pi r_{Ag}^3/3)\Delta T$, where $C_{Ag}$ is the heat capacitance of silver, $\rho_{Ag}$ is its density, $r_{Ag}$ is the cluster radius, $N$ is the number of clusters in the heated filament bottleneck, and $\Delta T$ is the temperature variation. Assuming $C_{Ag} = 0.236$ J/g °C, $\rho_{Ag} = 10.49$ g/cm$^3$, $N = 10$, $r_{Ag} = 10$ nm, and $\Delta T = 100$ °C, the required energy per oscillation is estimated to be 0.1 picoJ. This corresponds to a potential power consumption as low as $10^{-5}$ W at a signal cycle frequency of 0.1 GHz.

Transneurons sensitive to the relative phase of two signals could facilitate the creation of compact circuits capable of analysing

complex signals (see Section 12 in SI). AI systems built with transneurons may surpass current AI hardware, paving the way for dynamical cognitive processors[56]. For instance, estimating the distance to objects typically requires at least two conventional sensory neurons, which respond to slightly shifted signals (e.g. acoustical or optical), connected to a decision neuron that computes distance by comparing their activity. This three-neuron circuit could be replaced with a single transneuron that receives paired stimuli, functioning as a hybrid of sensor and decision neuron. Furthermore, tuning a transneuron to operate at the boundary between self-sustained and noise-induced regimes could enable advanced coding schemes, where stimulus information is encoded in the distribution of ISIs rather than solely in the ISI rate (see Section 10 in SI).

In conclusion, artificial neurons based on diffusive memristors can effectively emulate various types of biological neurons while supporting multiple coexisting spiking regimes within a single device. These features pave the way for future artificial intelligence systems with dynamical learning capabilities[56–58].

## Methods

### Physiological studies

#### Middle temporal cortical area (MT)

**Electrophysiological recordings.** Recordings were made in cortical areas MT in two adult male rhesus monkeys. Activity of single neurons was recorded using tungsten microelectrodes driven into cortex using a hydraulic micropositioner. Neurophysiological signals were filtered, sorted, and stored using the Plexon (Dallas, TX) system. Visual responses were recorded while the animals performed a fixation task. Firing rates were measured at five to seven different levels of luminance contrast in the vicinity of the neurons' preferred spatiotemporal frequencies. The different stimulus conditions and contrasts were interleaved in random order across trials. To compute population response (Figs. 5 and 6), the firing rates of individual neurons (ranging from 20 to 100 spikes/s) were normalised to their maximal firing rates[6]. See Section 13 of SI for additional recording details.

**Behavioural procedure.** Monkeys were seated in a standard primate chair with a surgically implanted head post rigidly affixed to the chair frame. The task was to fixate a small target in the presence of moving visual stimuli for the duration of each trial. The eye position was monitored and recorded. After eye position was maintained within a 2° window centred on the fixation target throughout the trial, animals were given a small juice reward. See Supplementary Materials for additional details of the procedure.

#### Parietal reach region (PRR)

**Electrophysiological recordings.** Single-unit recordings were made from both hemispheres in each of two adult male rhesus monkeys. Boundaries of cortical area PRR were identified based primarily on physiological criteria. See Section 13 of SI for additional recording details.

**Behavioural procedure.** Animals first fixated on a circular white stimulus centred on the screen in front of them. Left and right paws touched 'home' pads situated at waist height and in front of each shoulder. After holding the initial eye and hand positions, either one or two peripheral targets appeared on the screen. When two targets appeared, they were at opposite locations relative to the fixation point. After an additional period that lasted above 1 s, the central eye fixation target shrank in size to a single pixel, cueing the animal to move to the peripheral target(s) in accordance with target colour. Trials could be unimanual or bimanual. Animals were given a small juice reward on correct trials. See Supplementary Materials for additional detail of procedure.

**Motor cortex.** Neuronal recordings were performed in macaque monkeys performing a delayed reach-to-grasp task using Utah Arrays implanted along the central sulcus and overlapping the border between primary motor cortex and dorsal or ventral premotor cortex of the right hemispheres of two rhesus monkeys as described in ref. 40.

Experimental protocols were approved by the Animal Care and Use Committees of the Salk Institute and of the Washington University School of Medicine, and they conform to U.S. Department of Agriculture regulations and to the National Institutes of Health guidelines for the humane care and use of laboratory animals.

### Artificial neurons

**Fabrication.** *Method 1:* We fabricated the diffusive memristor devices used to obtained data for Figs. 3–7, S6, S8 on a p-type (100) Si wafer with 100 nm thermal oxide. The bottom electrodes were patterned by photolithography followed by evaporation and liftoff of ~20/2 nm Pt/Ti. The ~10 nm thick $SiO_2$ dielectric layer was deposited at room temperature by reactively sputtering $SiO_2$ in Ar. A 4 nm Ag layer was subsequently deposited using the same technique as the dielectric. The ~30 nm Pt top electrodes were subsequently patterned by photolithography followed by evaporation and liftoff processes. Electrical contact pads of the bottom electrodes were first patterned by photolithography and then subjected to reactive ion etching with mixed $CHF_3$ and $O_2$ gases.

*Method 2:* For measurements presented in Figs. 3, 5, 7, S2, and S6 the diffusive memristors were prepared by co-sputtering Ag and $SiO_2$ in Ar at room temperature. Dielectric layer of nominal thickness 40 nm deposited by co-sputtering was sandwiched between 30 (20) nm bottom (top) Pt electrodes. The top electrodes were sputtered through a shadow mask.

### Measurements

Electrical measurements described in Figs. 3–7 were performed using the Keysight 33622A arbitrary waveform generator, the Keysight MSOX3104 mixed signal oscilloscope, and the Keysight B1530 WGFMU. Voltage pulses were applied by the Keysight 33622A. The analogue oscilloscope channels were used to measure the voltages at the output of the function generator, and across the diffusive memristor. The current across the diffusive memristor was monitored using a 1.5 kΩ resistor connected in series to the diffusive memristor while the voltage across it was being monitored by one channel of the oscilloscope. We used electrolytic capacitors (100 nF) and general-purpose resistors (70 kΩ).

Electrical measurements presented in Figs. 3, 5 and 7 were performed using the Keithley 4200SCS, Rigol arbitrary waveform generator DG4162, and PicoScope 5443D oscilloscope on an Everbeing probe station. Electrical connections to the memristor were made via tungsten probe tips. Voltage pulses were applied by the waveform generator, and the oscilloscope channels were utilised to measure the voltage output of the pulse generator and the voltage across the diffusive memristor, which was hosted on a temperature-controlled chuck. The temperature of the chuck was controlled using an Everbeing temperature chuck controller and an ATC chiller. For the spiking circuit, we used a range of resistances between 55 and 90 kΩ general-purpose resistor and a 50 nF electrolytic capacitor.

### Simulations

To simulate spiking activities, we used the model, which describes the interplay of three key processes that govern charge transport: (i) Ag nanocluster diffusion in a dielectric medium, (ii) electron tunnelling between the terminals and the Ag clusters, and (iii) heat flow dynamics (see, e.g.[27,28,59]). The agreement between results of our simulations and measurements (both shown in Fig. 4) is evidence that the model captures the essential dynamics of the system and thus it can be used to analyse mechanisms of spiking.

Controlling spiking behaviour in artificial neurons requires understanding the relationship between stochastic and deterministic components in their dynamics. This task is particularly challenging for the diffusive memristor, where interactions between thermal, nano-mechanical and electrical degrees of freedom produce intriguing thermo-mechanical and thermo-electric dynamics coupled with the diffusive drift of Ag-clusters that form conducting filaments. The diffusion is controlled by the nanocluster temperature, which is in turn determined by a balance between Joule local heating generated by current through the memristor and cooling due to the heat sink to the substrate having bath temperature $T_0$ (Fig. 2B). The current and the corresponding voltage across the memristor is defined by Kirchhoff's circuit law (Fig. 2B), while the memristor resistance is governed by electron tunnelling through the Ag-clusters between memristor terminals.

For a nearly formed filament, the memristance is controlled by a few particles[60,61] in the gap between the filament tip and the electrode (Fig. S1B). In this case, the model equations read:

$$\eta \frac{dx_i}{dt} = -\frac{\partial U(x_i)}{\partial x_i} + q\frac{V}{L} - \sqrt{\eta D}\xi_i(t) \qquad (1a)$$

$$\frac{dT}{dt} = \frac{V^2}{C_{th}R(x_1, \ldots x_N)} - \kappa(T - T_0) \qquad (1b)$$

$$\tau \frac{dV}{dt} = V_{ext} - \left(1 + \frac{R_{ext}}{R(x_1, \ldots x_N)}\right)V + \sqrt{D_V}\varsigma_V(t). \qquad (1c)$$

Here, $x_i$ is the position of the $i$-th mobile particle (Ag-cluster) diffusing within the gap of size $L$, $\eta$ is the viscosity coefficient, $U$ is the phenomenological electro-chemical potential in the gap with a minimum near the top of the filament (Fig. S1A in SI), and $V$ is the voltage drop across the gap. We renumerate particles during simulations to always keep $x_i > x_{i-1}$. In this work, we consider the cases of either one or two Ag-particles in the gap (Fig. S1B in SI). In the case of one particle, the resistance of the memristor is

$$R(x_1) = R_t \left(e^{\frac{(L/2+x_1)}{\lambda}} + e^{\frac{(L/2-x_1)}{\lambda}}\right) = R(x) = R_0 \cosh\left(\frac{x}{\lambda}\right),$$

and for two particles it is

$$R(x_1, x_2) = R_t \left(e^{\frac{(L/2+x_1)}{\lambda}} + e^{\frac{(x_2-x_1)}{\lambda}} + e^{\frac{(L/2-x_2)}{\lambda}}\right).$$

Here $R_t$ is the tunnelling resistance amplitude, and $\lambda$ is the effective tunnelling length[59]. For one particle in the gap, the minimum resistance of the memristor $R_0 = 2R_t e^{L/2\lambda}$ occurs when the Ag-cluster is in the middle of the gap ($x = 0$). In the following we remove the subindex 1 in $x_1$ to simplify notation when modelling one Ag-particle in the gap. The Ag-particles are driven by an electrical force $qV/L$ with the induced effective cluster charge $q$; they diffuse inside the gap under the influence of a random force. In general, $q$ could depend on voltage and it can change its sign when voltage is inverted[61]. To avoid these unnecessary complications, we therefore consider only the case of $V_{ext} > 0$ and simplify the problem assuming a constant induced charge.

The noise affecting artificial neuron dynamics comes from two sources: (i) Brownian dynamics of Ag-clusters and (ii) external voltage noise in the artificial neuron circuits. These two sources of noise have counterparts in biological neurons: the noise due to diffusion of ions in ion channels and the noise in signals coming from other neurons in the neural network, connected to the given neuron. The noise due to diffusive dynamics of the Ag-clusters is modelled by Gaussian stochastic

forces $\xi_i(t)$ which has zero mean, are delta-correlated in time and statistically independent for each cluster. The intensity of random fluctuations is controlled by the diffusion constant, which is proportional to the temperature $D = 2k_B T$. The temperature is governed by Newton's cooling law, where the rate of heat transfer to the sink is determined by the cooling constant $\kappa$ and the background temperature $T_0$. We assume that the source of heat is the Joule dissipation, linked to temperature via the thermal capacitance $C_{th}$. We found that, in general, the model with two Ag-particles in the gap reproduces the current spiking observed in experiments with artificial neurons better than the model with one Ag cluster in the gap. However, the dynamics of the two-particles system is more difficult to interpret. Moreover, the broader distribution of $CV_2$ observed in the artificial neurons fabricated using Method 2, and having large values of $CV_1$, is better captured by the one-particle model. This observation suggests that the conductive pillars with one or two Ag clusters that control bottleneck resistance both exist in our diffusive memristors fabricated for this study. Here we use the two-particle model for the results presented in Fig. 4, and the one-particle model in other simulations.

Fluctuations of voltage mimicking the noise from other neurons appear in the equation (Eq. 1c), which describes the voltage drop $V$ across the memristor when an external voltage $V_{ext}(t)$ is applied. This equation is derived by applying the Kirchhoff current law to the circuit shown in Fig. 2B with the load resistance $R_{ext}$ and the circuit time constant $\tau = RC$. The neural network noise is added via random external voltage fluctuations $\sqrt{D_V}\varsigma_V(t)$, where $D_V$ is the intensity of voltage fluctuation and $\varsigma_V(t)$ is the uncorrelated Gaussian noise with zero mean.

The model (Eq. 1a–c) has been successfully used to describe stochastic dynamics of diffusive memristors[27–29]. Here we use it to analyse different spiking regimes. To distinguish between noise-induced and self-sustained spiking, we also develop a deterministic model (see justifications of this model in Section 1 in SI) where thermal fluctuations are replaced by additional deterministic forces related to the thermophoretic and/or thermoelectric effects[30,31]:

$$\eta \frac{dx}{dt} = -\frac{\partial U}{\partial x} - q\frac{V}{L} - q_T T', \qquad (2a)$$

$$\frac{dT'}{dt} = \frac{2VV'R - V^2R'}{C_{th}[R(x)]^2} - \kappa T', \qquad (2b)$$

$$\tau \frac{dV}{dt} = V_{ext} - \left(1 + \frac{R_{ext}}{R(x)}\right)V, \qquad (2c)$$

$$\tau \frac{dV'}{dt} = -\left(1 + \frac{R_{ext}}{R(x)}\right)V' + \frac{R_{ext}R'}{[R(x)]^2}V, \qquad (2d)$$

where $T'$ and $V'$ represent electric and thermal gradients in the gap, and we add a force proportional to $T'$ and a 'thermal charge' $q_T$ determined by the thermophoresis and/or Seebeck coefficients. Normalisations of the Eqs. (1) and (2) are discussed in Section 2 of SI.

## Reporting summary

Further information on research design is available in the Nature Portfolio Reporting Summary linked to this article.

## Data availability

All data needed to evaluate the conclusions of the study are present in the Main Manuscript and the Supplementary Information (no data were excluded). Data used for figures in the Main Manuscript and Supplementary Information, as well as raw data, are available at https://figshare.com/s/9869df87e8ab89d12bf3  https://doi.org/10.6084/m9.figshare.26310838.

## Code availability

Pascal codes used for simulations are provided at https://zenodo.org/records/15497230; https://doi.org/10.5281/zenodo.15497230. QtiPlot software were used (www.qtiplot.com) for statistical analysis.

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

## Acknowledgements

We would like to thank Prof. Qiangfei Xia for elucidating comments about the manuscript. We also thank the artist Irina Savelieva (irasavelieva.com) for creating an illustration used in the bottom panel of Fig. 1 and for granting the permission to use it under the CC BY 4.0 license. This work was supported by the Engineering and Physical Sciences Research Council grant No. EP/S032843/1 (S.E.S., P.B.), a National Eye Institute (NEI) Core Grant for Vision Research P30 EY019005 (T.D.A., S.G.), NEI grants 5R01EY012135 (L.H.S.), R01 EY018613 (T.D.A., S.G.), R01-EY029117 (T.D.A., S.G.), NIBIB + NIMH grant 1R01EB028154 (L.H.S.), NINDS R01 NS123435 (E.M.), the X-Grants Program of the President's Excellence Fund at Texas A&M University (R.S.W.), and Air Force Office of Scientific Research grant under contract no. FA9550-19-1-0213 (R.M., J.J.Y.).

## Author contributions

S.E.S. conceived the concept and coordinated writing the paper. S.E.S. and A.G.B. wrote the initial draft and performed stochastic (Eq. 1a–c) and deterministic (Eq. 2a–d) simulations. AGB performed bifurcation analysis of Eq. (2a–d) for artificial neuron. S.E.S., A.G.B., S.G. and R.S.W. wrote and revised the manuscript. S.S., J.J.Y. and A.G.B. designed the artificial neuron experiment and analysed the resulting data. S.G. led, and S.E.S. and A.G.B. contributed to the analysis of biological data. S.E.S., S.G., A.G.B., P.B., J.J.Y., T.D.A. and L.H.S. supervised experimental studies. R.M. and D.P. performed experiments on artificial neurons. A.S.P. and E.M. obtained data from MT and PRR biological neurons, respectively. All authors contributed to the discussion and editing of the manuscript. The first four authors contributed equally to the experimental studies reported in the manuscript.

## Competing interests

The authors declare no competing interests.

## Inclusion & Ethics

The authors have carefully considered researcher contributions and authorship criteria of multi-region collaboration to promote greater equity in this collaborative project. Experimental protocols for measurement of brain activity in macaque monkeys were approved by the Animal Care and Use Committee of the Salk Institute (MT data) and Institutional Animal Care and Use Committee of the Washington University (PRR data); these protocols conform to U.S. Department of Agriculture regulations and to the National Institutes of Health guidelines for the humane care and use of laboratory animals.
