## [Transparent Peer Review file · Nature Communications]

Artificial transneurons emulate neuronal activity in different areas of brain cortex

Corresponding Author: Professor Sergey Savel'ev

Version 0:

Reviewer comments:

Reviewer #1

(Remarks to the Author)

Review of the manuscript "Artificial transneurons emulate neuronal activity in different areas of brain cortex".

This paper combines the expertise of physicists, electrical engineers, and neuroscientists joined by the future goal of realizing some simple forms of artificial brains. For the first time, this paper reports about electronic devices capable to mimic spiking patterns of biological neurons from different areas of awaken monkey cortex. Switching between different patterns corresponding to visual, motor, and premotor neurons is controlled by the applied bias voltage, temperature, and external resistance. This is a transformative shift in neuromorphic technology since it provides a universal component for various neuromorphic circuits for brain-like computation and paves a path toward the future creation of very simple forms of artificial brains on a chip. The paper is quite systematic: presenting comprehensive measurements, simulations, and theoretical descriptions of fabricated artificial neurons and comparing the electrical responses with large-scale measurements of neural activities in awaken monkey brains. This work has the potential of having a significant influence in the interdisciplinary areas overlapping with physics, electronics and neuroscience, and, in principle, I recommend this work for publication in Nature Communications. However, I would like the author to consider my comments below before I come to a definite decision.

- 1) A reader could benefit from comparing simulated spiking patterns corresponding to different combinations of CV1-CV2 shown in Fig. 3A corresponding to different functional areas of monkey brain cortex.
- 2) In Figure 3A, it is evident that some point (CV1, CV2), corresponding to bursting in the living neurons, are not reproduced by artificial neurons. The authors should explain this discrepancy.
- 3) For the visual cortex, spiking depends on the stimulus intensity. Could the authors show how this affects the (CV1,CV2) distributions?
- 4) I believe that the work can benefit from more comprehensive introduction of various mem-devices in the context of the reported research.
- 5) Also, additional relevant references perhaps could be added in the Introduction, to place this work in a broader context.

Reviewer #2

(Remarks to the Author)

This paper explores stochastic firing behaviors of a memristor and find their similarities in different spiking activities of a biological neuron from different cortical regions, and the work further utilizes the memristor to perform brain-like computing. While memristors have been largely used to emulate synapses thus far and less frequently as spiking neurons, it is this second behavior that the authors focus on, going beyond what has been reported in this line of study. Since this paper demonstrates that memristors can be used not only as synapses but also as spiking neurons (with such bio-characteristic abilities as demonstrated in this paper), this work can motivate future efforts for more complex memristor-based integrated systems. Excellent work. Will appeal to a broad readership. At the same time, the paper is rather difficult to read, editorially (technical contents are dense, and sometimes I feel some backgrounds and connections would help reading), so I

recommend publication of this paper, after the authors consider putting reasonable efforts to make the reading easier of access. In connection with this, examples and clarifying questions are given below.

1. Stochastic behavior

When one considers a stochastic dynamics, there are two aspects to it. Source (what causes the stochastic behaviors), and the effect (what the source results in). Imagine, as well-known examples, an RC circuit, a Brownian motion, and an electronic clock linewidth. The stochastic effects are the voltage around the capacitor in the RC circuit (the kT/C deviation), the diffusion motion of the Brownian particle, and the clock line width or clock phase diffusion. The stochastic sources (origins) are electronic thermal motions (Nyquist noise) in the R, the random thermal motions of the molecules in the liquid where the Brownian particle is immersed, and the thermal, $1/f$, or shot noise of transistors and thermal noise of resistors in the electronic clock.

Now, in the stochastic dynamics the authors describe with the memristor and the biological neuron, what is the source? I understand the effect is, for example, the irregularity in the firing rate (is that correct?). I tried to understand this from the stochastic differential equation in the SI, but it was a bit difficult to follow. In the case of the memristor, is the source of the stochastic behavior something inherent/internal to the memristor (e.g., the way the conduction path is formed and ruptured)? (it appears that way, since a fixed voltage can give rise to the stochastic behaviors). In the case of the neuron, is the source of the stochastic behavior also internal to the neuron (like the memristor), or external to the neuron (like random electrochemical environment; i see the authors say "stochastic due to random chemical and electrical fluctuations of spiking in the neural tissue and the complex (and often uncontrollable) dynamics of neurons interacting in the network", but find it a bit difficult to follow, or at least not as explicit as I think it should be)? If the authors can shine light on this not just with equations but with understandable word descriptions, that would serve two purposes. First, better understanding of the work. Second, better understanding of whether the stochasticity of the memristor and that of the neuron have a similar structural analogy or are phenomenologically similar.

2 (rather minor). Figure 2 caption is difficult to read in the sense that subpanels cannot be understood immediately due to the way "a", "b"..... are scattered. A more traditional organizations would help. Up to the authors, but please at least consider.

3. (Partly mentioned in "1" above) I am trying to see whether the neuron-memristor similarity is due to the more fundamentally similar physical (or mathematical) structure, or phenomenological. For instance, the dynamics the authors cause with temperature (heating-cooling) might be seen biologically as well, but is it in the neuron also due to temperature? I think that in typical biological systems, temperature is quite tightly regulated and computing controlled by temperature would be deemed quite unstable and not viable through evolution.

4. Is it meaningful to ask how Fig. 2 is related to Fig. 3? In other words, the different regions (MT, PRR, etc.) in Fig. 3 have something to do with having particular fractions of combinations of Figs. 2A, B, and C dynamics? Since both Figs. 2 and 3 are described within the framework of stochasticity, providing such connection (if any) would help read the paper. At this point, if I read it correctly, they sound a bit disjoint.

5 (rather minor). By "The fact that transneurons can produce spiking with statistical characteristics different from biological cortical areas MT and PRR suggests....." the authors probably meant that they produce what is similar to MT and PRR and plus, something other than MT and PRR, correct? (The former was already stated in the foregoing part of the paper, so the above phrasing was a bit confusing).

6. Now, I see the value of what's presented in Figs. 4 and 5, and my question is, are these independent of, or directly related to, Figs. 2 and 3 (stochasticity)? I think such clarification would help the reader understand the flow of the paper story line.

7. (minor). Is "transneuron" a standard terminology? I felt a difficult word like that kind of blocks the paper reading, but there is no right answer to this. If the authors choose to use that word, probably articulate a bit on it? (just one sentence that explains the choice of that word?) In the current manuscript, it was rather done perfunctorily, e.g., "trans functional devices", etc.

Summary: This is an excellent work. Will attract attention from broad readership. I recommend publication enthusiastically. Clearer presentation as suggested above would make the paper more approachable.

Reviewer #3

(Remarks to the Author)

In the submitted manuscript, authors shows artificial transneurons that emulate neural activity in mammalian's brain. However, the advances from this work are not clear. The fabrication and theoretical analysis of diffusive memristor were already published by the numerous authors. Therefore, new findings or advances should be in emulating complex neuronal activity of the biological neurons in the brain cortex via artificial transneurons. However, the evaluation of transneurons in relation to the activity of biological neurons primarily focuses on the examination of spiking periodicity and their comparative analysis using the coefficient of variation. Since deterministic modeling of the diffusive memristor is not a new discovery, the authors had to explain what they accomplished by comparing biological activities in three distinct regions of the brain cortex with their artificial neurons. Unfortunately, the connections between the comparison and the demonstrated applications using transneurons are inadequate. Thus, the authors must provide a clear vision of how the transneuron can be used to

improve current artificial intelligence hardware. Listed below are several suggestions that would make the paper more suitable for Nature Communication.

1. According to the authors, the analysis of single neurons and their characteristics, known as receptive fields, is the first step in comprehending complex neural computations. (Line 32-33). However, it is difficult to comprehend the meaning of receptive fields in this text. This paper contains no additional description or explanation of the receptive field. In addition, there are no conclusive experiments to comprehend or simulate the receptive fields of biological neurons with artificial neurons.
2. In Lines 94-96, it is mentioned that spiking is suppressed at certain intermediate voltages. Is this phenomenon observed in biological neurons as well? It seems counterintuitive for a neuron to ignore stimuli within a specific range.
3. Fig. 2I and J (measured and simulated data, respectively) show that the spiking patterns gradually change from low voltage regimes to high voltage regimes. However, there is a discontinued regime from approximately 1.2 and 1.4 V in Fig. 2I (measured data), which does not correspond with Fig. 2J (simulation results). An adequate explanation is lacking in Lines 97-102.
4. In Fig. S3A-C, the Y-axis is labeled as "time stamps," ranging from 0 to 1. The intended meaning of this axis is unclear. It's possible it denotes a burst event.
5. Fig. 3 shows the comparison of biological and artificial neurons with analysis of CV1 and CV2 from spiking activity in MT, PRR, and PM. While the authors assert significant overlaps between PM and artificial transneurons in CV analysis, the extent of this overlap is debatable based on the visual data. There are numerous instances where the data points do not overlap. A quantitative analysis highlighting the degree of overlap would be beneficial.
6. In Fig. 3A, it is unclear if the measurement data follows the same trends depicted in the simulation data. It would be valuable if the authors could elucidate the reliability of the simulation data in comparison to the measured data.
7. Minor feedback: There are no (A) and (B) in Fig. 3.
8. In Lines 125-126, the relevance of the description to this study is ambiguous. It remains unclear if the demonstrated transneurons use the intermediate CV2 values for effective information processing. Further clarification on this point would be appreciated.
9. As this paper aims to understand the complex neural computations in biological neurons and compare them with the artificial neurons, it would be better to give more background on biological neurons' behavior or decision models in Line 186.
10. The selective activation of neurons is an important aspect of neural computation as the authors mentioned. I would appreciate a more detailed explanation or perspective on how this impacts efficient computation in biological neurons.
11. Minor feedback: A color bar beside Fig. 4C and D would be helpful.
12. What exactly is implied by 'information processing' in Line 200? Clarifying the specific function of the artificial neuron that changes during this type of information processing would enhance understanding.
13. For the phase detector using a single transneuron, it is difficult to find a connection to the understanding of biological neurons. It seems to be a separate application of the transneuron, as it utilizes the difference in phases of two input signals which have not been clearly mentioned in the biological neuron's aspects. It would be preferable to provide a clear path that can be used for routing information flow in complex neural systems in Line 216.
14. As neural network computing requires multiple neurons, I am curious if the authors can provide a scheme for arrays for future applications. Since the transneuron requires an external variable resistor or capacitor, the electrical lines to control might cause issues.
15. In Line 232, the authors mention implementing oscillations in bath temperature using a heating element attached to the neuron. Considering the significance of energy consumption in large-scale artificial neural network computing, it would be valuable if the authors could suggest methods to mitigate the potential energy consumption arising from these additional heating elements.
16. Line 236 may need more detailed applications to surpass the currently available AI hardware in terms of size or energy consumption.

Reviewer #4

(Remarks to the Author)

In the manuscript entitled "Artificial transneurons emulate neuronal activity in different areas of brain cortex" by Professor Savel'ev and colleagues, the authors presented the artificial transneurons based on diffusive memristors to emulate spiking

activity in different parts of biological neurons. Stochastic simulation of diffusive memristors is performed to reveal their distinct dynamical regimes, which fit well with the experiment results. This work provides sufficiently detailed simulations and extensive analysis of artificial neuron behavior. To meet the broad readership of Nature Communications, I suggest the authors to consider revising their manuscript according to the below comments.

Here are specific comments:

- 1) The two sub-figures in Figure 1 look uncorrelated, like two separated figures. Since the authors claim that a single artificial neuron can emulate different areas of brain cortex by tuning the applied voltage, bath temperature and circuit parameters, the correspondence between different parameters of artificial neurons and different biological brain areas should be clearly shown in Figure 1.
- 2) In page 2, line 71, the authors said "At low bias voltage, no spiking is observed. As voltage increases, the system first generates rare and random spikes", but neither the main text nor the supporting information (SI) shows the spiking activity mentioned or the detailed voltage parameters. The detailed voltage parameters of three different kinds of spiking activities shown in Fig. 2A, B, and C are also not mentioned.
- 3) Since this work points out the stochastic spiking behaviour, the cycle-to-cycle variation of the spiking activities should be measured. Is the stochastic spiking behaviour of this artificial neuron consistent each time when the same external voltage is applied and what is the endurance? In addition, only 3 kinds of spiking activities are shown in this manuscript, could the authors provide more measurement results under different voltage, bath temperature and circuit parameters?
- 4) Although this work emphasizes that artificial neurons can mimic different regions of biological neurons, only Figure 3 shows a comparison between the results of an artificial neuron and a biological neuron. Figures 2, 4, and 5 are all comparisons between the measurement results and the authors' simulation results of their artificial neuron.
- 5) In Figure 4C and D, why the dotted green curves show different changing trends? Does it mean that the measurement and simulation results show some disagreement with each other?
- 6) The quality of each figure should be improved. All of them are not clear. The sub-figures in Figure 3 are not labeled A and B. The label of each sub-figure should be placed outside the box of the figure to avoid overlap with the curves in the figure. Figure 5B is not mentioned in the main text. Each sub-figure is not the same size and is not aligned. The sub-figures in Figure 5 should be well-organized. It is hard to distinguish the simulation result and the measurement result.

Version 1:

Reviewer comments:

Reviewer #1

(Remarks to the Author)

Referee report.

manuscript entitled "Artificial transneurons emulate neuronal activity in different areas of brain cortex".

The initial manuscript was excellent and noteworthy, and this revised version is much better. The authors received a very large number of suggestions from four referees, and the authors did an excellent job addressing these issues. This revised manuscript is truly excellent, very well written, and presents a highly non-trivial very interdisciplinary study of very broad interest: can artificial transneurons emulate neuronal activity in different areas of brain cortex? This is a very interesting question, of very general interest to the broad readership of Nature Comm. I very strongly support the publication of this truly excellent work.

(Remarks on code availability)

It is very time consuming to check the code. Referees need to also do research.

Reviewer #2

(Remarks to the Author)

The authors have addressed most of my comments. I support the publication of this paper in Nature Communications.

(Remarks on code availability)

Reviewer #3

(Remarks to the Author)

The manuscript entitled "Artificial transneurons emulate neuronal activity in different areas of the brain cortex" has undergone substantial revision. I appreciate the authors' efforts in addressing the previous concerns, and many of the questions have been satisfactorily resolved. However, I remain skeptical about the claim that diffusive memristors can function as transneurons capable of emulating biological neurons. Despite the extended simulation results presented by the authors, I believe the most critical data are the measurement results. Unfortunately, the measured data still do not convincingly match the functions of biological neurons. Additionally, the figures are not helpful to understand the manuscript. I believe the revised manuscript is not suitable for publication in Nature Communications.

Comment 1: The paper aims to demonstrate the emulation of three different types of biological neurons—sensory, pre-motor, and motor neurons from a monkey—using artificial neurons. The key points should be: 1) The method by which the demonstrated device emulates the stochastic behavior of biological neurons; 2) How accurately the device emulates each type of biological neuron; 3) Its computational applications, showing the potential to function as a biological neuron or even surpass it with a smaller number of transneurons.

I believe the main issue lies in demonstrating how accurately the device emulates each type of biological neuron. In response to the original comments #5-6, the authors presented extended simulation data covering a broader range of biological neurons in the CV1 and CV2 space. However, the measured data still show discrepancies when compared with the simulation data and biological neuron data. The authors claim that the simulation and measured data show a clear, similar tendency in terms of CV1 and CV2 values. However, this claim is difficult to follow because the measured data do not exhibit a contour similar to that shown in the simulation data. Due to the limited amount of measured data, it seems the data are just spread out in CV1 and CV2 space. Without the simulation data points in the figure, the measurement data points would appear isolated from the biological neuron data. Only a few measured data points are close to the MT, PRR, and bursting data. It is difficult to claim that the fabricated memristor can function as a transneuron as proposed. The authors should provide more data to show the clear tendency. If it is not possible to improve the measurement data, it is hard to believe claim that the fabricated devices can function as transneurons, which make the manuscript unsuitable for publication.

Comment 2: I appreciate the authors' efforts to revise the manuscript with new figures. However, the laser spot shown in Fig. 1E is unrealistic. It might cause readers to mistakenly believe that the authors actually used a laser to control the device's temperature. Additionally, I doubt the existence of a commercially available laser that can focus on a few Ag clusters with such a small spot size, as depicted in Fig. 1E. It seems about 10-nm diameter spot size is drawn in the figure, which is challenging to achieve. While I understand that the authors wanted to show a potential method for controlling the bath temperature, I do not believe that heating the Ag clusters using a laser with such a small spot size is a generally acceptable idea.

Comment 3: The manuscript has been revised to include many supplementary equations and figures. However, these additions are scattered throughout the main manuscript, making it difficult to follow the overall flow. If the information in the supplementary material is essential, it should be integrated into the main manuscript. For example, in line 99, the text states, "As voltage increases, the system first generates rare and random spikes; it then evolves toward a regular spiking mode (see measurements and stochastic simulations in Figs. 2A and 2D, respectively, as well as Figs. 3C, S1D [green curve], S6C, and S9 in Supplementary Materials)." This scattered information prevents readers from grasping the main idea that the authors wish to convey.

Comment 4: In line 298, the manuscript states, "Simulations of the system using Eqs. (S3a)-(S3c) for VDC–VAC in the region of multistability (shown by the blue horizontal arrow in Fig. 5A) reveal a remarkable dependence of spiking intensity (Fig. 5B)." However, I did not find a clear description that provides evidence of a "remarkable dependence" of spiking intensity in Fig. 5B. I suggest that the authors rewrite the manuscript with fewer subjective adjectives and adverbs.

Comment 5: Unfortunately, the revised manuscript is difficult to read and understand. I believe the manuscript lacks sufficient interpretation of the findings and the associated figures. For instance, from line 278, the authors describe how input voltage governs the "transition" of transneurons between dynamical types. However, the description is inadequate for understanding how this transition occurs as a function of input voltage. The sentence "The trajectory of $x(t, V_{ext}(t))$ exhibits hysteresis behavior (dynamical multistability) as the voltage V_{ext} first increases and then decreases" reads more like a figure caption than an explanation of how input voltage governs the transition. The authors should explain what causes this hysteresis and when it occurs during the input voltage sweep in detail.

Comment 6: The caption for Fig. S1B, which relates to Fig. 5A, refers to "B at top" and "B at bottom." However, there is no top and bottom in the figure.

Comment 7: The caption for Figure 5 is not well organized. Each figure should have a proper caption with clearly defined labels.

Minor comment: I believe the data is the most important aspect of the manuscript. However, considering that the manuscript was submitted to Nature Communications, the quality of the figures is not adequate. This is not about the aesthetics of the figures, but rather their ability to convey information. Especially, Figure 1 does not effectively guide the readers to understand what the authors have done and what they aim to present.

(Remarks on code availability)

Reviewer #4

(Remarks to the Author)

The authors revised the manuscript, but this article still has several key issues:

The supplementary material (SI) lacks sufficient data support. While the authors mention adding new figures to the SI, you

are unable to access this material, and the descriptions do not indicate that a substantial amount of actual data is provided. Figure 1 fails to effectively demonstrate the purported "strong connection between the monkey brain and their device." This critical result requires clearer presentation and explanation.

In the main figures, only Figure 3 compares the measurement data and neuron data, but even here the authors acknowledge that the measurement data cannot strictly reproduce the biological neuron data characteristics. Additionally, the authors state that the amount of biological data is limited.

The other data is also not presented in a well-organized manner.

Overall, this article lacks sufficient data support and clear presentation, and requires further refinement to better substantiate the claims made.

(Remarks on code availability)

Version 2:

Reviewer comments:

Reviewer #3

(Remarks to the Author)

Authors address all the raised issue, so I recommend its publication in nature communications.

(Remarks on code availability)

Reviewer #4

(Remarks to the Author)

The authors have made satisfactory revisions. I would recommend the acceptance.

(Remarks on code availability)

REVIEWER COMMENTS

Reviewer #1 (Remarks to the Author):

Review of the manuscript "Artificial transneurons emulate neuronal activity in different areas of brain cortex".

This paper combines the expertise of physicists, electrical engineers, and neuroscientists joined by the future goal of realizing some simple forms of artificial brains. For the first time, this paper reports about electronic devices capable to mimic spiking patterns of biological neurons from different areas of awoken monkey cortex. Switching between different patterns corresponding to visual, motor, and premotor neurons is controlled by the applied bias voltage, temperature, and external resistance. This is a transformative shift in neuromorphic technology since it provides a universal component for various neuromorphic circuits for brain-like computation and paves a path toward the future creation of very simple forms of artificial brains on a chip. The paper is quite systematic: presenting comprehensive measurements, simulations, and theoretical descriptions of fabricated artificial neurons and comparing the electrical responses with large-scale measurements of neural activities in awoken monkey brains. This work has the potential of having a significant influence in the interdisciplinary areas overlapping with physics, electronics and neuroscience, and, in principle, I recommend this work for publication in Nature Communications. However, I would like the author to consider my comments below before I come to a definite decision.

We are glad that the reviewer recognized the value of our interdisciplinary study. We appreciate the positive comments and constructive suggestions.

1) A reader could benefit from comparing simulated spiking patterns corresponding to different combinations of CV₁-CV₂ shown in Fig. 3A corresponding to different functional areas of monkey brain cortex.

Following this suggestion, we have added new panels C-E to Figure 3, which displays examples of typical simulated spiking with CV₁-CV₂ characteristics corresponding to different cortical areas: MT, PRR, and PM (detailed in the caption). In addition, we provided examples of simulated and measured transneuron spiking in Figs. S2, S6C-E, S9-11.

2) In Figure 3A, it is evident that some point (CV₁, CV₂), corresponding to bursting in the living neurons, are not reproduced by artificial neurons. The authors should explain this discrepancy.

Because of the limited amount of biological data and our ability to study only a subset of parameters in the simulated data, there are data points from biological measurements that fall outside of the region of the data simulated for artificial transneurons. Following the helpful suggestion of Reviewer 2, now we estimate the degree of overlap of the simulated (in a model with two simulated particles in the bottleneck) and biological data. The overlap is as follows: 52% for PRR, 96% for MT, and 62% for PM (see the discussion in Supplementary Materials on pages 10-12, section "Overlapping estimates"). As seen in Supplementary Materials (Figure S2B-C), trains of bursting vary in terms of noise intensity (i.e., the amount of noise, Eqs. (S3), which is a dimensionless version of Eqs. (S1)), indicating that biological bursting data that fall outside of the region of the overlap can be covered by simulations in a noisier environment (please see our response to Reviewer 2) or to simulations with different numbers of particles in the bottleneck gap (e.g., see new simulations shown in Fig. S6B where the overlap of simulated data with biological data and measurements is considerably greater).

3) For the visual cortex, spiking depends on the stimulus intensity. Could the authors show how this affects the (CV₁, CV₂) distributions?

Following this suggestion, we have performed a new analysis of the data recorded from biological neurons in cortical area MT. Now the CV₁-CV₂ probability distributions for low stimulus contrast and high stimulus contrast are presented in a new Figure S5 and described in a new section in Supplementary Materials. The new figure shows that distributions of CV₁-CV₂ for different simulation intensities significantly overlap, supporting the notion that the MT data for all stimulus contrasts can be presented as one cloud of points in Fig. 3A. As expected, we demonstrate in Fig. S5 that the parameter CV₂ decreases with increasing contrast, in agreement with the idea that synchronisation of spiking by external signal is responsible for neuronal selectivity.

4) I believe that the work can benefit from more comprehensive introduction of various mem-devices in the context of the reported research.

To address this comment, we have expanded the introductions and added more references to help the reader appreciate the extensive literature on memristive devices, including the literature on quantum and classical memristive systems.

5) Also, additional relevant references perhaps could be added in the Introduction, to place this work in a broader context.

We have added additional references to make the article more accessible to the broad readership of Nature Communications.

We thank the reviewer again for presenting a positive assessment our work and for their constructive criticism.

Reviewer #2 (Remarks to the Author):

This paper explores stochastic firing behaviors of a memristor and find their similarities in different spiking activities of a biological neuron from different cortical regions, and the work further utilizes the memristor to perform brain-like computing. While memristors have been largely used to emulate synapses thus far and less frequently as spiking neurons, it is this second behavior that the authors focus on, going beyond what has been reported in this line of study. Since this paper demonstrates that memristors can be used not only as synapses but also as spiking neurons (with such bio-characteristic abilities as demonstrated in this paper), this work can motivate future efforts for more complex memristor-based integrated systems. Excellent work. Will appeal to a broad readership. At the same time, the paper is rather difficult to read, editorially (technical contents are dense, and sometimes I feel some backgrounds and connections would help reading), so I recommend publication of this paper, after the authors consider putting reasonable efforts to make the reading easier of access. In connection with this, examples and clarifying questions are given below.

We appreciate the reviewer's positive evaluation of our study and are very grateful for the helpful suggestions to improve the readability of our paper.

1. Stochastic behavior

When one considers a stochastic dynamics, there are two aspects to it. Source (what causes the stochastic behaviours), and the effect (what the source results in). Imagine, as well-known examples, an RC circuit, a Brownian motion, and an electronic clock linewidth. The stochastic effects are the voltage around the capacitor in the RC circuit (the kT/C deviation), the diffusion motion of the

Brownian particle, and the clock line width or clock phase diffusion. The stochastic sources (origins) are electronic thermal motions (Nyquist noise) in the R, the random thermal motions of the molecules in the liquid where the Brownian particle is immersed, and the thermal, $1/f$, or shot noise of transistors and thermal noise of resistors in the electronic clock.

Now, in the stochastic dynamics the authors describe with the memristor and the biological neuron, what is the source? I understand the effect is, for example, the irregularity in the firing rate (is that correct?). I tried to understand this from the stochastic differential equation in the SI, but it was a bit difficult to follow. In the case of the memristor, is the source of the stochastic behavior something inherent/internal to the memristor (e.g., the way the conduction path is formed and ruptured)? (it appears that way, since a fixed voltage can give rise to the stochastic behaviors). In the case of the neuron, is the source of the stochastic behaviour also internal to the neuron (like the memristor), or external to the neuron (like random electrochemical environment; i see the authors say "stochastic due to random chemical and electrical fluctuations of spiking in the neural tissue and the complex (and often uncontrollable) dynamics of neurons interacting in the network", but find it a bit difficult to follow, or at least not as explicit as I think it should be)? If the authors can shine light on this not just with equations but with understandable word descriptions, that would serve two purposes. First, better understanding of the work. Second, better understanding of whether the stochasticity of the memristor and that of the neuron have a similar structural analogy or are phenomenologically similar.

In addressing the above comment, we have revised the text where we consider two types of noise in our stochastic equations, with an additional term representing voltage fluctuations, Eq. (S1c). One source of noise represented in Eq. (S1a) as a random force, can be attributed to the intrinsic noise due to a stochastic process in a single neuron. This intrinsic noise mimics the noise in ion channels of biological neurons associated with diffusive charge transport. This type of noise is emulated by a random force acting on diffusive Ag-clusters, which we considered in detail in the original version of the manuscript. Additionally, we now consider another type of fluctuation that occurs due to noisy inputs from other neurons, which can be mimicked by the noise of external voltage. Thus, in the present model, the stochastic equations describe both internal and external noises.

We have also performed extensive simulations to study how these noises influence the relationship between CV_1 and CV_2 characteristics, as shown in Fig. S1F-G, for different values of voltage noise intensity and bath temperature (or memristor substrate temperature). Our results indicate that noise from either source has a greater effect on the regular spiking mode (similar to PRR spiking) compared to the already noisy spiking regimes (similar to the spiking activity in MT and PM areas). For example, noise can push the distribution of CV_1 - CV_2 for transneurons toward the edge of the PRR spiking region. We also observe (in Fig. S1F) that both types of noise have similar effects on stochasticity (discussed in the Supplement in the section "Internal (diffusive) and external (network) noises and their influence on system dynamics"). Thus, we believe that our results are robust against both types of noise.

2. (rather minor). Figure 2 caption is difficult to read in the sense that subpanels cannot be understood immediately due to the way "a", "b"..... are scattered. A more traditional organizations would help. Up to the authors, but please at least consider.

We have rearranged the panels and labels.

3. (Partly mentioned in "1" above) I am trying to see whether the neuron-memristor similarity is due to the more fundamentally similar physical (or mathematical) structure, or phenomenological. For instance, the dynamics the authors cause with temperature (heating-cooling) might be seen biologically as well, but is it in the neuron also due to temperature? I think that in typical biological

systems, temperature is quite tightly regulated and computing controlled by temperature would be deemed quite unstable and not viable through evolution.

In artificial neurons, heating-cooling cycles modulate the diffusion constant of Ag-clusters due to temperature variations (since, in our formalism, the diffusion constant is proportional to Ag-cluster temperature). We agree with the reviewer that significant temperature modulations cannot occur in biological neurons. However, changes of diffusion constants can certainly affect the dynamics of biological neurons. Therefore, we believe that the analogy between heating-cooling cycles in artificial neurons and spiking variations observed in biological neurons is valid due to the modulation of ion diffusion in ion channels [Amir et al., Oscillatory mechanism in primary sensory neurones, *Brain* 125, 421 (2002); Siegelbaum & Tsien, Modulation of gated ion channels as a mode of transmitter action, *Trends in Neurosciences* 6, 307 (1983)]. In biological neurons, diffusion variations could arise from factors such as changes in channel shape, influenced by varying concentrations of ATP or oxygen in the neurons' environment. These concentrations are in turn affected by neuron's spiking activity: as ATP and oxygen are consumed during neuronal activity, their levels decrease, which slows down the spiking activity. Conversely, reduced spiking allows ATP and oxygen levels to increase, initiating the next cycle of spiking rate oscillation.

4. Is it meaningful to ask how Fig. 2 is related to Fig. 3? In other words, the different regions (MT, PRR, etc.) in Fig. 3 have something to do with having particular fractions of combinations of Figs. 2A, B, and C dynamics? Since both Figs. 2 and 3 are described within the framework of stochasticity, providing such connection (if any) would help read the paper. At this point, if I read it correctly, they sound a bit disjoint.

The spiking activity shown in Fig. 2 (regimes A/D and C/F) corresponds to the PRR-like and bursting regimes in the artificial neuron. To address this question in detail, we have provided examples of simulated spiking that correspond to different (CV_1 , CV_2) regions in panels C-E of Fig. 3, and in Fig. S6B-E, where different spiking regimes are depicted across various areas of the CV_1 - CV_2 space.

5. (rather minor). By "The fact that transneurons can produce spiking with statistical characteristics different from biological cortical areas MT and PRR suggests....." the authors probably meant that they produce what is similar to MT and PRR and plus, something other than MT and PRR, correct? (The former was already stated in the foregoing part of the paper, so the above phrasing was a bit confusing).

We thank the reviewer for sharing this observation. We agree that this sentence was rather misleading and we have rephrased it.

6. Now, I see the value of what's presented in Figs. 4 and 5, and my question is, are these independent of, or directly related to, Figs. 2 and 3 (stochasticity)? I think such clarification would help the reader understand the flow of the paper story line.

To address this question, in addition to the already shown sharpening of ISI distributions for the "natural" AC period, we have studied how stimulus selectivity manifests itself in the relationship between spiking rate (the number of spikes per unit of time) and the period of AC-voltage. Performing this analysis was motivated by the fact that, for biological neurons, stimulus selectivity is defined in terms of the maximal spiking rate elicited at a certain magnitude of the stimulus: such as the temporal frequency of visual stimuli (see new Fig. 4E for MT neurons). For artificial transneurons, we also observe a maximum spiking rate at a certain AC-period (shown for experimental data in the new Fig. 4F and for simulations in the new Fig. 4G). Notice that the period at which spiking rate reaches its maximum shifts toward low values as stimulation increases: AC-voltage amplitude increases for transneurons (Fig. 4G) or stimulus contrast increase for MT neurons (Fig. 4E). This observation

indicates that transneurons can reproduce nonlinear behaviour of biological neurons (often referred to as “non-classical” receptive fields).

Next, we performed additional simulations to study whether (and how) stimulus selectivity occurs at higher levels of transneuron stochasticity. In particular, we simulated ISI distributions as a function of AC-voltage period for artificial neurons (Fig. S7A) with the same simulation parameters as in Fig. 4 but with the DC voltage corresponding to Fig. 2F. These distributions are dominated by short ISIs with no time scale, but they still exhibit a weak peak at a long ISI (which is characterised by the “natural” ISI time scale) visible in Fig. S7A as a white blob. This maximum in ISI distributions is affected by the period of AC-voltage. That is, when the period of the applied AC-voltage is comparable to the natural ISI time scale, the maximum of the ISI distribution becomes more pronounced (represented by a brighter-white spot), and it is elongated in the direction of the yellow dashed line (which represents the conditions, at which the period of stimulation is equal to the “natural” time of the neuron). This occurs in full analogy to the less stochastic case illustrated in Fig. 4D. This result is a manifestation of stimulus selectivity for bursting transneurons. We also plot the maximum rate of the bursting transneuron as a function of the AC-period. The maximum rate shifts toward lower AC-periods (see Fig. S7B) with increasing AC-voltage amplitude (which mimics stimulus intensity), again in full analogy with stimulus selectivity of regular transneuron spiking (shown in Fig. 4G) and MT spiking (Fig. 4E).

The above discussion demonstrates how spiking with different stochasticity described in Figs. 2 and 3 affects computational ability of the transneuron discussed in Fig. 4, thus providing a link requested by the reviewer.

7. (minor). Is "transneuron" a standard terminology? I felt a difficult word like that kind of blocks the paper reading, but there is no right answer to this. If the authors choose to use that word, probably articulate a bit on it? (just one sentence that explains the choice of that word?) In the current manuscript, it was rather done perfunctorily, e.g., "trans functional devices", etc.

We use the term “transneuron” to indicate that the same physical device – an artificial neuron – can transit between spiking characteristics of biological neurons of different types. We believe that the capability for transition between different kinds of neurons is aptly captured by this term. This discussion is summarised in the main manuscript (lines 85-87).

Summary: This is an excellent work. Will attract attention from broad readership. I recommend publication enthusiastically. Clearer presentation as suggested above would make the paper more approachable.

We thank the reviewer both for their most useful comments and for expressing a positive opinion about our work.

Reviewer #3 (Remarks to the Author):

In the submitted manuscript, authors shows artificial transneurons that emulate neural activity in mammalian's brain. However, the advances from this work are not clear. The fabrication and theoretical analysis of diffusive memristor were already published by the numerous authors. Therefore, new findings or advances should be in emulating complex neuronal activity of the biological neurons in the brain cortex via artificial transneurons. However, the evaluation of transneurons in relation to the activity of biological neurons primarily focuses on the examination of spiking periodicity and their comparative analysis using the coefficient of variation. Since deterministic modeling of the diffusive memristor is not a new discovery, the authors had to explain what they accomplished by comparing biological activities in three distinct regions of the brain cortex with their artificial neurons. Unfortunately, the connections between the comparison and the demonstrated applications using transneurons are inadequate. Thus, the authors must provide a clear vision of how the transneuron

can be used to improve current artificial intelligence hardware. Listed below are several suggestions that would make the paper more suitable for Nature Communication.

We appreciate the reviewer's insightful feedback and valuable suggestions to improve our manuscript. As noted by all other reviewers, a notable achievement of our study is the demonstration of an artificial neuron that faithfully reproduces the complex stochastic behaviour of biological neuron in the monkey brain. These findings pave the way for developing advanced artificial intelligence hardware capable of faithfully emulating brain functions, going beyond current hardware that only simulates rudimentary processes. Such advancements not only deepen our understanding of the brain but also offer powerful computational tool with the potential to approach natural intelligence.

1. According to the authors, the analysis of single neurons and their characteristics, known as receptive fields, is the first step in comprehending complex neural computations. (Line 32-33). However, it is difficult to comprehend the meaning of receptive fields in this text. This paper contains no additional description or explanation of the receptive field. In addition, there are no conclusive experiments to comprehend or simulate the receptive fields of biological neurons with artificial neurons.

The concept of "receptive field" in neurophysiology characterizes the responses of individual neurons, yet acknowledges that these responses depend on properties of the larger network—an ensemble of interacting neurons (further discussed in [6]). In physics, a similar concept known as "mean field" describes an effective field produced by other particles and acting on a representative particle. This approach proved useful for analysis of systems with many interacting particles when the exact description is intractable. We discuss this analogy in the revised manuscript's introduction. Figs. 4C-D, E, G and S7A-B illustrate various characteristics of measured and simulated receptive fields. Additionally, Fig. 1B-C illustrates how the concept of receptive field is used to interpret neuronal selectivity.

To demonstrate similarity in the responses between transneurons and various neurons within the brain's neural network of awake monkey, we plotted simulated and measured (experimental) rates of transneurons against AC-voltage periods in Figs. 4F-G and S7B. We have compared the responses of transneurons and biological MT neurons (Fig. 4E-G). Specifically, we depicted the spiking rates of transneurons as a function of AC period and voltage amplitude. For MT neurons, we analysed spiking rates in response to temporal periods of visual stimuli at different stimulus intensities (contrasts). In both cases, the rates peak at specific periods of stimulus modulation, consistent with the notions of neuronal selectivity and receptive field. These comparisons are illustrated in the colour plots of transneuron activity in the bottom panels of Figs. 4E, G and S7B.

2. In Lines 94-96, it is mentioned that spiking is suppressed at certain intermediate voltages. Is this phenomenon observed in biological neurons as well? It seems counterintuitive for a neuron to ignore stimuli within a specific range.

Physiological studies demonstrate that only stimulus within the neuron receptive field can elicit spiking above the spontaneous level. Consequently, infrequent spiking in noise-induced modes can be interpreted as suppressed activity when the stimulus falls outside of the transneuron's receptive field. By applying sufficiently strong AC voltage, spiking in these noise-induced regimes can be elicited in transneurons (Fig. S8). Furthermore, the dependence of transneuron spiking on AC amplitude at the boundary between noise-induced and self-sustained regimes can be used for "beyond rate" coding (Fig. S8).

Biological neurons exhibit both spontaneous and stimulus-evoked spiking. In neurophysiological studies, stimulus-evoked spiking is recorded when the stimulus parameters, such as location and/or characteristics of spatial and temporal frequencies of luminance modulation, are within the receptive

field. Prior to conducting our physiological experiments, we systematically mapped the receptive field of the neurons under study. This mapping process entailed identifying specific stimulus parameters that corresponded to the receptive fields of the neurons and evoked spiking responses above spontaneous activity levels. Subsequently, all stimuli were presented within the spatial region of each neuron's receptive field.

3. Figures 2I and J (measured and simulated data, respectively) show that spiking patterns gradually change from low-voltage to high-voltage regimes. However, there is a discontinued regime from approximately 1.2 and 1.4 V in Fig. 2I (measured data), which does not correspond with Fig. 2J (simulation results). An adequate explanation is lacking in Lines 97-102.

The "intensity" of sparse spiking in the noise-induced regime noted by the Reviewer depends on the filament structure, heat capacitance and heat-sink parameters of the transneuron. These parameters, influence whether noise-induced spiking is more or less frequent. To illustrate this variability, we have included a new figure (Fig. S1E in the Supplementary Materials), which depicts an instance of sparse spiking occurring between self-sustained spiking modes. This scenario arises when two clusters within the bottleneck (gap) have slightly different charges (due to, e.g., slightly different size of Ag-clusters, resulting in partial cancellation of noise).

4. In Fig. S3A-C, the Y-axis is labeled as "time stamps," ranging from 0 to 1. The intended meaning of this axis is unclear. It's possible it denotes a burst event.

The timestamps shown in Fig. S3A-C illustrate a common method of representing spiking activity in the neuroscientific literature. This method is employed when researchers focus solely on the spiking rate of neurons without considering the shape of the neuronal signals. In this case, a spike is recorded when the signal exceeds a pre-defined threshold, thereby reducing the neural response to a series of timestamps (Fig. S3).

5. Fig. 3 shows the comparison of biological and artificial neurons with analysis of CV1 and CV2 from spiking activity in MT, PRR, and PM. While the authors assert significant overlaps between PM and artificial transneurons in CV analysis, the extent of this overlap is debatable based on the visual data. There are numerous instances where the data points do not overlap. A quantitative analysis highlighting the degree of overlap would be beneficial.

Per reviewer request, we have created a new figure (Fig. S6A in Supplementary Materials) in which we quantified the overlap of CV₁-CV₂ characteristics of transneuron and biological neurons in cortical areas MT, PRR, and PM. To this end, we first derived a contour (shown in grey) that contains 99% of the data (grey hexagons) in the (CV₁, CV₂) plot of simulated spiking activity of transneurons. Then we computed the percentage of data points falling inside this contour for MT, PRR, and PM neural populations. We would like to note that the region of the 99% contour can be expanded, leading to a larger magnitude of the overlap in several ways, for example (a) using higher bath temperature and by adding noise to the external voltage, as noted in our response to Reviewer 2, or (b) simulating a different number of diffusing Ag-clusters in the bottleneck (as shown in Fig. S6B).

6. In Fig. 3A, it is unclear if the measurement data follows the same trends depicted in the simulation data. It would be valuable if the authors could elucidate the reliability of the simulation data in comparison to the measured data.

The spiking sequences obtained in our experiments are much shorter than the simulated sequences, resulting in that the measured data are noisier than the simulated data. However, there is a clear tendency for simulated spiking at higher voltages toward higher values of CV₁, consistent with the measured data (Fig 3A). The wider spread of CV₂ values in the measured data can be explained by a higher noise intensity or a stronger dependence of resistance on Ag-cluster dynamics. As a first step

toward addressing this issue, we performed simulations of the transneuron with one Ag-cluster in a bottleneck resulting in a sharper $R(x)$ -dependence (compared to the two-cluster simulations presented in Fig. 3). The results reveal a broadening of the distribution of CV_2 values for larger CV_1 values, similar to our measurements of transneurons (now reported in the Supplement; Fig. S6B).

7. Minor feedback: There are no (A) and (B) in Fig. 3.

We thank the reviewer for pointing this out. We have corrected the error in the revised manuscript.

8. In Lines 125-126, the relevance of the description to this study is ambiguous. It remains unclear if the demonstrated transneurons use the intermediate CV_2 values for effective information processing. Further clarification on this point would be appreciated.

We show that the transneuron can be tuned to attain the values of CV_2 between 0.5 and 1.5 for the two-cluster model (Fig. 3A) or even between 0.1 and 1.9 for one cluster model (Fig. S6B). This possibility indicates the potential for significant optimisation of information processing by transneurons. However, a study on optimal information processing is beyond the scope of this work and will require (i) extension of the work to many interconnected neurons so one can model information transfer from node to node, and (ii) a decision-making protocol to simulate how many spikes the system needs to receive before it can make a decision. This discussion is summarised in the main manuscript.

9. As this paper aims to understand the complex neural computations in biological neurons and compare them with the artificial neurons, it would be better to give more background on biological neurons' behavior or decision models in Line 186.

In the revised manuscript we address this issue in the subsection "Transneural computations."

10. The selective activation of neurons is an important aspect of neural computation as the authors mentioned. I would appreciate a more detailed explanation or perspective on how this impacts efficient computation in biological neurons.

We have provided a more detailed explanation of our perspective on neural computation in the revised manuscript in the introduction and in subsections "Transneurons emulate visual, motor and pre-motor neurons in the monkey brain" and "Computation by Selectivity".

Selectivity is a crucial function of neurons, essential for analysing sensory signals (see also our response to Comment 12 from the same reviewer below). For example, the selectivity manifests itself by peaks in the neuron's spiking rate, in response to specific stimulus features. Updated Fig. 1 elucidates how selectivity contributes to neural information processing and decision making. Additionally, selectivity is illustrated in Figs. 4 and S7. In Fig. 4E the rate of MT neurons is plotted against stimulus oscillation period at various stimulus contrasts. For comparison, Fig. 4F shows the measured rate of transneuron as a function of AC voltage period. Figs. 4G and S7B portray simulated rates of a transneuron against different AC voltage period at different AC voltage amplitudes and different degrees of stochasticity. In every case, a peak in spiking rate occurs at specific temporal periods, highlighting sensitivity to particular stimulus frequencies (selectivity). Interestingly, we observe a shift of the maximum rate towards low values of the stimulation period (Fig. 4E for MT, Fig. 4G for transneurons in low stochastic mode, and Fig. S6B for transneurons in high stochastic mode). This indicates that the nonlinear features of artificial transneurons and biological neurons are similar.

11. Minor feedback: A colour bar beside Fig. 4C and D would be helpful.

We have added a colour bar to these panels.

12. What exactly is implied by 'information processing' in Line 200? Clarifying the specific function of the artificial neuron that changes during this type of information processing would enhance understanding.

The revised Fig. 1 provides a general explanation of information processing in biological systems, focusing on analogue signal processing rather than logic gate. This approach is similar to the computation of binocular disparity (underlying stereoscopic vision) and motion parallax (underlying motion perception) in biological vision, as discussed in the revised introduction and subsection "Transneuronal computations." Signal comparison is a critical function in living organisms; for instance, comparing relative phase of signals (i.e., relative delay) from two ears or eyes helps to determine the direction or distance to an object. This computation should involve at least three artificial neurons (as in Fig. S12): two neurons receiving stimuli and one making decisions. A single transneuron can replace such a network by hybridising sensory and decision-making features (Fig. S12B). A circuit comprising two transneurons and one artificial neuron can handle multimodal information (e.g., visual and auditory) from four channels for comparison (as shown in Fig. S12C). Furthermore, tuning a transneuron to the boundary between noise-induced and self-sustained regimes enables beyond-rate information coding and processing (as mentioned above and outlined in Fig. S8).

13. For the phase detector using a single transneuron, it is difficult to find a connection to the understanding of biological neurons. It seems to be a separate application of the transneuron, as it utilizes the difference in phases of two input signals which have not been clearly mentioned in the biological neuron's aspects. It would be preferable to provide a clear path that can be used for routing information flow in complex neural systems in Line 216.

To address this concern, we added a discussion of how biological neurons can compare signals: in the first paragraph of the subsection "Computation by signal comparison."

14. As neural network computing requires multiple neurons, I am curious if the authors can provide a scheme for arrays for future applications. Since the transneuron requires an external variable resistor or capacitor, the electrical lines to control might cause issues.

To respond to this question, we have created a new diagram presented in Fig. S12. The external capacitor and resistor can be associated with resistance of memristor terminals and self-capacitance of the memristor. We can control these parameters during the process of fabrication (Fig. S12). Indeed, we can fabricate memristors with larger-area terminals (thus affecting memristor capacitance) and deposit additional material to change contact resistance. Therefore, there is no need in additional elements to realise transneurons. In this case, the tuning will be realised via applied voltage and additional noise (as in Figs. 1I-J, S1F-G, S4, S9-10 allowing to tune transneuron to PRR, PM, and MT like spiking), thus without any additional fabrication to change neuron type. We have also proposed a possible fabrication procedure.

15. In Line 232, the authors mention implementing oscillations in bath temperature using a heating element attached to the neuron. Considering the significance of energy consumption in large-scale artificial neural network computing, it would be valuable if the authors could suggest methods to mitigate the potential energy consumption arising from these additional heating elements.

Per reviewer request, we provide potential ideas for mitigation of how to minimise consumption of energy for temperature variation. This can be done, e.g., by means of laser pulse (Fig. 1) or local electrical heater. Estimates of power consumption are added at the end of the section “Phase detection mechanism” in Supplementary Materials.

16. Line 236 may need more detailed applications to surpass the currently available AI hardware in terms of size or energy consumption.

We have included an example demonstrating how transneurons can enhance signal comparison capabilities at the end of the paragraph just before the conclusion. Additionally, in the section "Small Functional Neuromorphic Circuits with Transneurons" in the Supplementary Materials, we show that transneurons can perform the same tasks using fewer neurons (see also Fig. S7). This highlights their potential for more complex, multimodal information processing.

Reviewer #4 (Remarks to the Author):

In the manuscript entitled “Artificial transneurons emulate neuronal activity in different areas of brain cortex” by Professor Savel’ev and colleagues, the authors presented the artificial transneurons based on diffusive memristors to emulate spiking activity in different parts of biological neurons. Stochastic simulation of diffusive memristors is performed to reveal their distinct dynamical regimes, which fit well with the experiment results. This work provides sufficiently detailed simulations and extensive analysis of artificial neuron behavior. To meet the broad readership of Nature Communications, I suggest the authors to consider revising their manuscript according to the below comments.

We thank the reviewer for the positive evaluation and for providing us with useful suggestions on how to improve the manuscript for the broad readership of Nature Communications.

Here are specific comments:

1) The two sub-figures in Figure 1 look uncorrelated, like two separated figures. Since the authors claim that a single artificial neuron can emulate different areas of brain cortex by tuning the applied voltage, bath temperature and circuit parameters, the correspondence between different parameters of artificial neurons and different biological brain areas should be clearly shown in Figure 1.

We have redesigned Figure 1, and we rewrote the related text to help the reader better understand connections between different parts of this work.

2) In page 2, line 71, the authors said “At low bias voltage, no spiking is observed. As voltage increases, the system first generates rare and random spikes”, but neither the main text nor the supporting information (SI) shows the spiking activity mentioned or the detailed voltage parameters. The detailed voltage parameters of three different kinds of spiking activities shown in Fig. 2A, B, and C are also not mentioned.

We have now provided multiple examples of measured spiking of transneurons at different voltages and temperatures, in Figs S9-S10. For example, one can see from Fig. S9 that the PRR-like spiking has been observed within certain voltage range (from 0.6V and 1.3V) for the samples fabricated by using method 1 (see Supplemental Materials). At applied voltage of 0.6V, the spiking is rare and quite irregular (Fig S9a). The PRR-like spiking develops for voltages 0.7 – 1V and then degrades and disappeared in a good agreement with simulations Fig. S1E. For bursting regimes of neurons fabricated by method 2 (see Methods in the main manuscript), spiking occurs between 1V and 1.9V (see Fig. S10a-e). For MT-like spiking in the samples fabricated by method 2, spiking starts at about 1.2V and the average ISI gradually decreases with increasing DC voltage (Fig. 10f-h).

3) Since this work points out the stochastic spiking behaviour, the cycle-to-cycle variation of the spiking activities should be measured. Is the stochastic spiking behaviour of this artificial neuron consistent each time when the same external voltage is applied and what is the endurance? In addition, only 3 kinds of spiking activities are shown in this manuscript, could the authors provide more measurement results under different voltage, bath temperature and circuit parameters?

Now we have presented results of repetitive measurements of the same transneuron driven by three sequential voltage pulses (Fig. S11A). The measured spiking differs from pulse to pulse (S11B-C). That is, the spikes triggered by each pulse do not coincide with one another (do not occur at the same time moments from the beginning of each pulse), but the overall spiking stochastic dynamics excited by these three pulses is very similar. We have also presented the evolution of PRR-like, MT-like, and bursting transneuron when voltage changes on Figs. S9, S10 (A-H).

As we have shown earlier (*Physical Review Applied* 19 [2], 024065, 2023), temperature can significantly affect spiking of artificial neurons based on diffusive memristors, including resetting sample to high resistive state if it is stuck in a low resistive state, thus improving endurance of artificial neurons. Here we have provided an example (Fig. S10I-J) in which we show, in an experiment, that a moderate increase of temperature increases the frequency of spiking.

4) Although this work emphasizes that artificial neurons can mimic different regions of biological neurons, only Figure 3 shows a comparison between the results of an artificial neuron and a biological neuron. Figures 2, 4, and 5 are all comparisons between the measurement results and the authors' simulation results of their artificial neuron.

We have added two new figures (Figs. 4E, S5) that describe measuring selectivity in biological neurons and comparing it with artificial neurons, along with examining the influence of stimulus intensity on spiking activity. Also, the original version of supplementary materials had a figure (now Fig. S3A) showing biological bursting timestamps and spiking histograms (now Fig S8), in addition to Fig. 3 mentioned by the reviewer.

5) In Figure 4C and D, why the dotted green curves show different changing trends? Does it mean that the measurement and simulation results show some disagreement with each other?

To prevent sample degradation, the number of spikes was deliberately kept significantly lower in measurements than simulations. Consequently, the dependence of the maximum probable ISI on the period of AC-voltage oscillations observed in experiments (with both increasing and decreasing segments) is noisier than that in simulations. Nonetheless, a consistent trend emerges where the maximum probable ISI drops at higher AC periods, mirroring the observed trend in simulations.

6) The quality of each figure should be improved. All of them are not clear. The sub-figures in Figure 3 are not labelled A and B. The label of each sub-figure should be placed outside the box of the figure to avoid overlap with the curves in the figure. Figure 5B is not mentioned in the main text. Each sub-figure is not the same size and is not aligned. The sub-figures in Figure 5 should be well-organized. It is hard to distinguish the simulation result and the measurement result.

We have improved the quality of figures per request of the reviewer and editors.

Responses to Reviewers' comments

Reviewer #1:

Reviewer's comment

This revised manuscript is truly excellent, very well written, and presents a highly non-trivial very interdisciplinary study of very broad interest: can artificial transneurons emulate neuronal activity in different areas of brain cortex? This is a very interesting question, of very general interest to the broad readership of Nature Comm. I very strongly support the publication of this truly excellent work.

Response

We would like to thank the reviewer for their kind comments and the recommendation to publish our work.

Reviewer #2 (Remarks to the Author)

Reviewer's comment

The authors have addressed most of my comments. I support the publication of this paper in Nature Communications.

Response

We would like to thank the reviewer for their recommendation to publish this paper in Nature Communications.

Reviewer #3 (Remarks to the Author)

Reviewer's comment

I appreciate the authors' efforts in addressing the previous concerns, and many of the questions have been satisfactorily resolved.

Response

We would like to thank the reviewer for appreciating the volume of work required to address the comments of all the reviewers, which took more than half a year of additional measurements and data analyses.

Reviewer's comment

However, I remain skeptical about the claim that diffusive memristors can function as transneurons capable of emulating biological neurons. Despite the extended simulation results presented by the authors, I believe the most critical data are the measurement results. Unfortunately, the measured data still do not convincingly match the functions of biological neurons.

Response

In the first round of revision, we provided a significant amount of additional simulations, and we expanded our measurements and analyses of data. But we understand that the measurements of CV_1 - CV_2 characteristics could still be

seen as insufficient to create a convincing picture of simulation trends. Now we have increased the number of measured points by the factor of about 14, from 116 to 1613. As discussed below, now we have shown that the “hammer-like” pattern of measured data closely follows the pattern of simulated data and, more importantly, the transneuron measurements significantly overlap with the biological data. In particular, we have demonstrated that the measured CV₁-CV₂ data well overlap with the biological results from different areas of the monkey brain cortex. To estimate the percentage of stochastic characteristics of biological neurons from different cortical areas that can be mimicked by transneurons, we provide overlap estimates of the measured (CV₁, CV₂) values for artificial neurons and PRR/MT/PM neurons in Fig. S6.

Reviewer’s comment:

I believe the main issue lies in demonstrating how accurately the device emulates each type of biological neuron. In response to the original comments #5-6, the authors presented extended simulation data covering a broader range of biological neurons in the CV₁ and CV₂ space. However, the measured data still show discrepancies when compared with the simulation data and biological neuron data. The authors claim that the simulation and measured data show a clear, similar tendency in terms of CV₁ and CV₂ values. However, this claim is difficult to follow because the measured data do not exhibit a contour similar to that shown in the simulation data. Due to the limited amount of measured data, it seems the data are just spread out in CV₁ and CV₂ space. Without the simulation data points in the figure, the measurement data points would appear isolated from the biological neuron data. Only a few measured data points are close to the MT, PRR, and bursting data. It is difficult to claim that the fabricated memristor can function as a transneuron as proposed. The authors should provide more data to show the clear tendency.

Response

As noted above, we have increased the number of measured CV₁-CV₂ points in Fig. 5 by the factor of 14. First, the presented measured data well overlap with simulated data and thus they follow the “hammer-like” shape of the CV₁-CV₂ cloud of the simulated data. The only region where the measured and simulated data do not overlap (as simulated pattern area are slightly bigger than measured one) is at high CV₂ and low CV₁ values. This region is difficult to reach experimentally with this specific technology due to the unavoidable formation of multiple pillars in fabricated memristors. Such multipillar structures lead to higher noise and prevents spiking at lower CV₁ when CV₂ is high. However, this region can be reached by diffusive memristors with artificial defects (e.g., see [10.1039/D3NR01853A](https://doi.org/10.1039/D3NR01853A)). Additionally, the CV₁-CV₂ points simulated with two Ag clusters cover a smaller area, suggesting that the absence of experimental CV₁-CV₂ points in the high CV₂, low CV₁ region may be due to the presence of multiple mobile clusters in the fabricated transneurons. Second, and more importantly, the measured CV₁-CV₂ points of the transneuron overlap the CV₁-CV₂ points for the biological data from MT, PRR, and bursting neurons (70% for PRR, 100% for MT, and 100% for PM cortical neurons; see Fig. S6 in the SI).

We have also demonstrated how accurately the transneuron can emulate the activity of biological neurons. For example, Fig. 5A1 shows that, within a certain voltage range, the CV_1 - CV_2 data for the same transneuron fit within the CV_1 - CV_2 region of MT neurons. Therefore, in this parameter range, all measurements demonstrated that the transneuron's stochasticity matched that of MT neurons. Additionally, Fig. 3A2 experimentally shows that the same artificial neuron, with a fixed load resistance, can shift its stochastic characteristics from those typical of PRR neurons to those of bursting neurons by increasing the applied voltage. Approximately half of the CV_1 - CV_2 points fall within the PRR region, while the other half fall within the bursting activity region.

Reviewer's Comment

I appreciate the authors' efforts to revise the manuscript with new figures. However, the laser spot shown in Fig. 1E is unrealistic. It might cause readers to mistakenly believe that the authors actually used a laser to control the device's temperature. Additionally, I doubt the existence of a commercially available laser that can focus on a few Ag clusters with such a small spot size, as depicted in Fig. 1E. It seems about 10-nm diameter spot size is drawn in the figure, which is challenging to achieve. While I understand that the authors wanted to show a potential method for controlling the bath temperature, I do not believe that heating the Ag clusters using a laser with such a small spot size is a generally acceptable idea.

Response

We agree that achieving laser heating on a 10 nm scale is challenging with standard technology. For this reason, we have removed the laser spot from the figure mentioned by the reviewer (now Fig. 2C).

In the discussion, we consider another local heating technique: a nanoscale Joule heater, as described in a recent article cited in the manuscript. Specifically, e-beam lithography and thin-film deposition could be used to design a thin-film heater on a 100 nm scale, providing electrical Joule heating on top of our device. Nevertheless, a laser with a near-field lens remains a possible option for achieving the necessary local heating. For this reason, we included a brief discussion of this technique and provided relevant references, demonstrating a lateral resolution of 10-50 nm for such systems.

A new discussion of different local heater options has been added to the main text of the manuscript.

Reviewer's comment

The manuscript has been revised to include many supplementary equations and figures. However, these additions are scattered throughout the main manuscript, making it difficult to follow the overall flow. If the information in the supplementary material is essential, it should be integrated into the main manuscript. For example, in line 99, the text states, "As voltage increases, the system first generates rare and random spikes; it then evolves toward a regular spiking mode (see measurements and stochastic simulations in Figs. 2A and 2D, respectively, as well as Figs. 3C, S1D [green

curve], S6C, and S9 in Supplementary Materials).” This scattered information prevents readers from grasping the main idea that the authors wish to convey.

Response

We had significantly expanded supplementary materials to address the reviewers’ comments during the first rounds of revision. This included providing additional evidence requested by Reviewers 1 and 2, both of whom have now recommended the manuscript for publication. To further assist readers in following our article, we have now reduced the number of direct references to Supplementary Information (SI) in the main text. We have also moved one figure (illustrating experimental results that show the evolution of spiking regimes with changing voltage and temperature) and the main equations (along with their justification) from SI to the main manuscript; they now appear in the Method section. Additionally, we have transferred most of the text where we interpret our results and the physical bifurcation mechanisms of spiking from SI to the main text.

Reviewer’s comment

In line 298, the manuscript states, “Simulations of the system using Eqs. (S3a)-(S3c) for VDC–VAC in the region of multistability (shown by the blue horizontal arrow in Fig. 5A) reveal a remarkable dependence of spiking intensity (Fig. 5B).” However, I did not find a clear description that provides evidence of a “remarkable dependence” of spiking intensity in Fig. 5B. I suggest that the authors rewrite the manuscript with fewer subjective adjectives and adverbs.

Response

We agree with the reviewer that the manuscript contained an excessive use of emphatic adjectives and adverbs. In the revised text, we have removed many of these to improve clarity and conciseness.

Reviewer’s comment

Unfortunately, the revised manuscript is difficult to read and understand. I believe the manuscript lacks sufficient interpretation of the findings and the associated figures. For instance, from line 278, the authors describe how input voltage governs the “transition” of transneurons between dynamical types. However, the description is inadequate for understanding how this transition occurs as a function of input voltage. The sentence “The trajectory of $x(t, V_{\text{ext}}(t))$ exhibits hysteresis behaviour (dynamical multistability) as the voltage V_{ext} first increases and then decreases” reads more like a figure caption than an explanation of how input voltage governs the transition. The authors should explain what causes this hysteresis and when it occurs during the input voltage sweep in detail.

Response:

The mechanisms of metastability are based on the bifurcation analysis, which was previously described in Supplementary Information. Following the reviewer’s suggestion, we have moved this content to the main text and expanded the corresponding discussion and interpretation, enabling readers to

more readily appreciate the complex dynamics of artificial neurons. We have also provided an explanation of dynamical hysteresis requested by the reviewer.

Reviewer comment

The caption for Fig. S1B, which relates to Fig. 5A, refers to “B at top” and “B at bottom.” However, there is no top and bottom in the figure.

Response

We thank the reviewer to noticing this inconsistency. We have now revised the text accordingly.

Reviewer’s comment

The caption for Figure 5 is not well organized. Each figure should have a proper caption with clearly defined labels.

Response

We have revised the caption of that figure (now Fig. 7) to clearly specify which portions of the caption correspond to each panel of the figure.

Reviewer’ comment

I believe the data is the most important aspect of the manuscript. However, considering that the manuscript was submitted to Nature Communications, the quality of the figures is not adequate. This is not about the aesthetics of the figures, but rather their ability to convey information. Especially, Figure 1 does not effectively guide the readers to understand what the authors have done and what they aim to present.

Response

We have reorganized the materials presented in the former Fig. 1, by separately presenting the concept of neuronal selectivity and the concept of transneuron, respectively in Figs. 1 and 2.

Reviewer #4 (Remarks to the Author)

The authors revised the manuscript, but this article still has several key issues:

Reviewer’ comment

The supplementary material (SI) lacks sufficient data support. While the authors mention adding new figures to the SI, you are unable to access this material, and the descriptions do not indicate that a substantial amount of actual data is provided.

Response

This comment may have arisen due to miscommunication. All supplementary figures should have been available to reviewers, as we uploaded them during submission. The data for supplementary figures were provided in a separate zip file, uploaded alongside all other files. To prevent further miscommunication,

we have now included a README file that describes all the data accompanying the revised manuscript.

Reviewer's comment

Figure 1 fails to effectively demonstrate the purported "strong connection between the monkey brain and their device." This critical result requires clearer presentation and explanation.

Response

Fig. 1 originally served as an introductory illustration to the concept of the transneuron. To clarify matters, we have now split it into two figures. The new Fig. 1 illustrates the concept of neuronal selectivity, while the new Fig. 2 introduces the concept of the transneuron.

Reviewer Comments

In the main figures, only Figure 3 compares the measurement data and neuron data, but even here the authors acknowledge that the measurement data cannot strictly reproduce the biological neuron data characteristics. Additionally, the authors state that the amount of biological data is limited.

Response

In the new revision, we have provided a significantly more measured CV₁-CV₂ data points for the transneuron, increasing the amount of data ~14-fold. In the newly revised manuscript, we clearly demonstrate a significant overlap between the measured biological data and the measured transneuron data, thus experimentally validating the concept of the transneuron (Fig. S6). The quantities of both biological and transneuron measurements are now sufficient for validation, with a statistically significant number of measured points overlapping across all the studied areas of the biological cortex.

Note that the biological measurements, including the comparison of measured biological and measured transneuron spiking characteristics, are now presented in Figs. 2, 5-6 of the main manuscript, and in Figs. S3, S5-6, and S8 of Supplementary Information.

Reviewer's comment

The other data is also not presented in a well-organized manner. Overall, this article lacks sufficient data support and clear presentation, and requires further refinement to better substantiate the claims made.

Response

As noted above, we have now provided about 14 times more measured CV₁-CV₂ data points than before. All shared data are described in the accompanying README file for the revised manuscript. For completeness, we provide the README content below.

README :

Data are provided in a zip archive submitted with the manuscript. The following are descriptions of all the shared files.

----- **Fig 2 data** -----

Examples of time stamps of the spikes measured in monkey cortex and in the transneurons at different voltages and circuit parameters are available in the folder \Data\Fig2\

- Three files contain examples of biological data for cortical areas MT, PRR and PM, called respectively MT.txt, PRR.txt and PM.txt.
- Three files contain examples of transneuron data in the parameter regions where transneurons emulate the corresponding cortical areas, Trans-MT.txt, Trans-PRR.txt and Trans-PM.txt.
- The file *plot_spiketrain_demos.m* contains the MATLAB code that reads and plots the illustrative spike trains.

----- **Fig 3 data** -----

Measured transneuron spiking data of the current $I(t)$ and/or voltage $V(t)$ across the memristor are available in the folder: \Data\Fig3\.

The voltage and current are given in Volts and Amperes, respectively, and time t is given in seconds. The values of external resistances and capacitances, as well as the information about fabrication methods, are presented in the caption of Fig. 3.

I) Transneuron fabricated by Method 1:

Measured transneuron $V(t)$ and $I(t)$ for $V_{\text{ext}}=0.6\text{V}$:
 \Data\Fig3\Fig3a.txt

Measured transneuron $V(t)$ and $I(t)$ for $V_{\text{ext}}=0.7\text{V}$:
 \Data\Fig3\Fig3b.txt

Measured transneuron $V(t)$ and $I(t)$ for $V_{\text{ext}}=1\text{V}$:
 \Data\Fig3\Fig3c.txt

Measured transneuron $V(t)$ and $I(t)$ for $V_{\text{ext}}=1.1\text{V}$:
 \Data\Fig3\Fig3d.txt

Measured transneuron $V(t)$ and $I(t)$ for $V_{\text{ext}}=1.3\text{V}$:
 \Data\Fig3\Fig3e.txt

II) Transneurons fabricated by Method 2:

Measured transneuron #1 $V(t)$ for $V_{\text{ext}}=1\text{V}$:
 \Data\Fig3\Fig3f.txt

Measured transneuron #1 $V(t)$ for $V_{\text{ext}}=1.3\text{V}$:
 \Data\Fig3\Fig3g.txt

Measured transneuron #1 $V(t)$ for $V_{\text{ext}}=1.4\text{V}$:
 \Data\Fig3\Fig3h.txt

Measured transneuron #1 $V(t)$ for $V_{\text{ext}}=1.5\text{V}$:
 \Data\Fig3\Fig3i.txt

Measured transneuron #1 $V(t)$ for $V_{\text{ext}}=1.9\text{V}$:
 \Data\Fig3\Fig3j.txt

Measured transneuron #2 $V(t)$ for $V_{\text{ext}}=1.2\text{V}$:
 \Data\Fig3\Fig3k.txt

Measured transneuron #2 $V(t)$ for $V_{\text{ext}}=2.2\text{V}$:
 \Data\Fig3\Fig3l.txt

Measured transneuron #2 $V(t)$ for $V_{\text{ext}}=3\text{V}$:
 \Data\Fig3\Fig3m.txt

Measured transneuron #3 $V(t)$ for $T_0= 20\text{C}^0$, $V_{\text{ext}}=3\text{V}$:
 \Data\Fig3\Fig3m.txt

Measured transneuron #3 $V(t)$ for $T_0= 40\text{C}^0$, $V_{\text{ext}}=3\text{V}$:
 \Data\Fig3\Fig3m.txt

----- **Fig 4 data** -----

All measurements were performed at the room temperature, for samples fabricated by Method 1. Simulation parameters and simulation units are described in Table ST1 and in Section 2 of Supplementary Information. Additional information about measurements is available in the caption of Fig 4 and in the data files, which are collected in the folder \Data\Fig4\.

The measured temporal response of the transneuron is given in mA/V - the ratio of the measured current and the applied external voltage $I(t)/V_{\text{ext}}$, as a function of time t given in ms, for $V_{\text{ext}}=0.7\text{V}$:
 \Data\Fig4\Fig4A-0_7V.txt

The measured temporal response of the transneuron is given in mA/V - the ratio of the measured current and the applied external voltage $I(t)/V_{\text{ext}}$, as a function of time t given in ms, for $V_{\text{ext}}=1.4\text{V}$:
 \Data\Fig4\Fig4B-1_4V.txt

The measured temporal response of the transneuron is given in mA/V - the ratio of the measured current and the applied external voltage $I(t)/V_{\text{ext}}$, as a function of time t given in ms, for $V_{\text{ext}}=1.8\text{V}$:
 \Data\Fig4\Fig4C-1_8V.txt

The simulated temporal response of the transneuron is given as the ratio of the current and the applied external voltage $I(t)/V_{\text{ext}}$ normalised by R_{min} as a function of time t normalised by τ for dimensionless $V_{\text{ext}}=10$:
 \Data\Fig4\Fig4D-10.txt

The simulated temporal response of the transneuron is given as the ratio of the current and the applied external voltage $I(t)/V_{\text{ext}}$ normalised by R_{min} as a function of time t normalised by τ for dimensionless $V_{\text{ext}}=32$: \Data\Fig4\Fig4E-32.txt

The simulated temporal response of the transneuron is given as the ratio of the current and the applied external voltage $I(t)/V_{\text{ext}}$ normalised by R_{min} as a function of time t normalised by τ for dimensionless $V_{\text{ext}}=195$: \Data\Fig4\Fig4F-195.txt

Power density spectrum obtained from measured transneuron response $I(t)/V_{\text{ext}}$ at $V_{\text{ext}}=0.7\text{V}$:
 \Data\Fig4\Fig4G-0_7V.txt

Power density spectrum obtained from measured $I(t)$ for sample fabricated by Method 1 at $V_{\text{ext}}=1.4\text{V}$:
 \Data\Fig4\Fig4G-1_4V.txt

Power density spectrum obtained from measured $I(t)$ for sample fabricated by Method 1 at $V_{\text{ext}}=1.8\text{V}$: \Data\Fig4\Fig4G-1_8V.txt

Power density spectrum obtained from simulated $I(t)$ at $V_{\text{ext}}=10$:
 \Data\Fig4\Fig4H-10.txt

Power density spectrum obtained from simulated $I(t)$ at $V_{\text{ext}}=32$:
 \Data\Fig4\Fig4H-32.txt

Power density spectrum obtained from simulated $I(t)$ at $V_{\text{ext}}=195$:
 \Data\Fig4\Fig4H-195.txt

Slow voltage sweep ($dV_{\text{ext}}/dt=1\text{V}/100$ sec) performed to collect measured transneuron response $I(V_{\text{ext}}(t))/V_{\text{ext}}(t)$, given in mA/V:
 \Data\Fig4\Fig4I.txt

Slow voltage sweep ($dV_{\text{ext}}/dt=1.4*10^{(-3)}$ normalised units) performed to collect simulated transneuron response
 $R_{\text{min}}I(V_{\text{ext}}(t))/V_{\text{ext}}(t)$:
 \Data\Fig4\Fig4J.txt

----- **Fig 5 data** -----

Comparison of the statistics of spiking for CV1-CV2 characteristics (see main text) in three different areas of monkey cortex, as well as in measured and simulated transneurons.

CV1-CV2 obtained from measured timestamps for MT area of monkeys' cortex:
 \Data\Fig5\Fig5A_MT

CV1-CV2 obtained from measured timestamps for PM area of monkeys' cortex:
 \Data\Fig5\Fig5A_PM.txt

CV1-CV2 obtained from measured timestamps for PRR area of monkeys' cortex:
 \Data\Fig5\Fig5A_PRR.txt

CV1-CV2 obtained from the transneuron's timestamps, measured for voltages 0-3V, external resistances $30\text{k}\Omega < R_{\text{ext}} < 90\text{k}\Omega$, external capacitances $1\text{nF} < C < 100\text{nF}$:
 \Data\Fig5\Fig5A-transneuron-measurements.txt

CV1-CV2 obtained from simulated transneuron timestamps for voltages $0 < V_{\text{ext}}/V_{\text{th2}} < 9$, the external resistances $0 < G_{\text{max}}R_{\text{ext}} < 5000$, and capacitances $0.006 < C < 30$:
 \Data\Fig5\Fig5A-grey-transneuron-simulations.txt

CV1-CV2 obtained from the transneuron measured for external voltages 0.6V-0.8V, external resistance $R_{\text{ext}}=65\text{k}\Omega$ and capacitance 1nF:
 \Data\Fig5\Fig5A1-transneuron-measurements.txt

CV1-CV2 obtained from transneuron timestamps measured for external voltages 1.06V-1.22V, external resistance $R_{\text{ext}}=70\text{k}\Omega$, capacitance $C=100\text{nF}$:

\Data\Fig5\Fig5A2-transneuron-measurements.txt

CV1 as a function of the external resistance and voltage obtained from simulated transneuron timestamps for voltages $0 < V_{\text{ext}} / V_{\text{th2}} < 9$, $1 < G_{\text{max}} R_{\text{ext}} < 5000$, and capacitance $C=30/R_{\text{ext}}$:

\Data\Fig5\Fig5B.txt

Simulated $I(t)$ for PRR-like (\Data\Fig5\Fig5C.txt), PM-like (\Data\Fig5\Fig5D.txt), and MT-like (\Data\Fig5\Fig5E.txt) regimes of the transneuron, with the simulation parameters listed in Table ST1, and the values of CV1, CV2 described in the caption of Fig. 5d.

----- **Fig 6 data** -----

Comparison of selectivity of biological neurons and transneurons. Parameters of measurement are provided in the figure caption, while simulation parameters are presented in the caption of Fig.6, in Table ST1, and in Section 2 of Supplementary Information.

Applied external voltage (given in V) versus time given in sec (\Data\Fig6\Fig6A_V.txt), and measured current response (spikes) in Amperes versus time given in sec: \Data\Fig6\Fig6A_I.txt.

Applied external voltage (with both AC and DC components and normalised on DC voltage) versus time (\Data\Fig6\Fig6B_V.txt), and simulated current response (spikes) normalised by $G_{\text{max}} V_{\text{DC}}$ versus time given in units of 'RC' time tau: \Data\Fig6\Fig6B_I.txt.

Measured distributions of inter-spike intervals (ISIs) as a function of the period of applied AC voltage given in ms: \Data\Fig6\Fig6C.txt.

Simulated distributions of inter-spike intervals (ISIs) as a function of the period of applied AC voltage (ISI measured in 'RC' time tau): \Data\Fig6\Fig6D.txt.

Measured spiking rate for monkey cortical area MT stimulated by a Gabor stimulus as a function of normalised contrast and time period given in sec:

\Data\Fig6\Fig6E.txt

Measured transneuron spiking rate (spikes/s) as a function of the AC voltage period (in ms) for DC voltage of 1V and AC voltage of 0.5V, $R_{\text{ext}}=70\text{k}\Omega$, $C=100\text{nF}$:

\Data\Fig6\Fig6F.txt

Simulated transneuron spiking rate (spikes/\tau) as a function of AC voltage amplitude, $V_{\text{AC}}/V_{\text{DC}}$ (i.e., normalised by DC voltage), at a fixed DC voltage and oscillation period t_p/τ (see Fig. 6 caption and Table ST1 for detail):

\Data\Fig6\Fig6F

----- **Fig 7 data** -----

Fig.7 contains both simulated and measured data. Details of simulations and simulation parameters are presented in the caption of Fig. 7, Table

ST1, and in Section 2 of SI. For measured data, voltage is given in V, temperature in degrees Celsius (normalised by 45 °C), time in seconds.

Ag cluster diffusion (position x of one cluster in the bottleneck versus time) simulated using stochastic (Eq. 1a-c) and deterministic (Eq. 2a-d) equations, respectively:

\Data\Fig7\Fig7FA_stoch.txt and \Data\Fig7\Fig7A_determ.txt.

The simulated spiking rate is presented as a 2D contour plot of two variables: (i) the AC voltage amplitude, normalized by the DC voltage (V_{DC}), and (ii) the phase difference between two signals—one being encoded in voltage and the other being encoded in temperature:

\Data\Fig7\Fig7B.txt

Simulated relation of conductance spikes normalised by its maximum value (G_{max}) versus time normalised by τ_{ae} given in \Data\Fig7\Fig7C.txt for in-phase signals and in \Data\Fig7\Fig7D.txt for anti-phase signals. The system is driven by two signals, which are AC voltage oscillations normalised by V_{DC} and temperature oscillations T normalised by T_{DC} as a function of time normalised by τ given in \Data\Fig7\Fig7E.txt for in-phase oscillations and in \Data\Fig7\Fig7E.txt for anti-phase oscillations.

Measured transneuron response of voltage across the diffusive memristor (normalised by the instant applied external voltage) versus time: \Data\Fig7\Fig7G.txt. The transneuron is driven by two in-phase signals: AC voltage vs time and temperature vs time. These signals are available in \Data\Fig7\Fig7I_V.txt and \Data\Fig7\Fig7I_T.txt, respectively, with time given in sec.

Measured transneuron response of voltage across the diffusive memristor (normalised by the instant applied external voltage) versus time: \Data\Fig7\Fig7H.txt. The transneuron is driven by two anti-phase signals: AC voltage vs time and temperature vs time. These signals are available in \Data\Fig7\Fig7J_V.txt and \Data\Fig7\Fig7J_T.txt, respectively, with time given in sec.

----- **Fig S1 data** -----

Details of simulations and simulation parameters can be found in the caption of Fig. S1, Table ST1, and in Section 2 of SI.

Dimensionless potential energy used to calculate dU/dx in simulations (see Eq. 1a and Eq. 2a, see Section 2 of SI):

\Data\FigS1\FigS1A-pot.txt.

Ag-cluster diffusion (position x versus time) simulated using deterministic equations (Eq. 2a-d) at external applied voltages, where the system exhibits: (i) self-sustain spiking due to charging-discharging oscillations (\Data\FigS1\FigS1C-green.txt), (ii) attracted to fixed points (\Data\FigS1\FigS1C-red.txt and \Data\FigS1\FigS1C-magenta.txt), and (iii) more complex self-sustained oscillations due to the heating-cooling cycle (\Data\FigS1\FigS1C-orange.txt).

Ag-cluster diffusion (position x versus time) simulated using stochastic equations (Eq. 1a-c) at external applied voltages, where the system exhibits (i) self-sustain spiking due to charging-discharging oscillations (\Data\FigS1\FigS1D-green.txt), (ii) fluctuations near a

fixed point (\Data\FigS1\FigS1D-red.txt), and (iii) more complex self-sustained oscillations due to the heating-cooling cycle (\Data\FigS1\FigS1D-orange.txt).

Simulated current $I(t)$ normalised by $R_{\min}V_{\text{ext}}(t)$ versus external voltage $V_{\text{ext}}(t)$ in units of threshold voltage V_{th2} , when the external voltage slowly varies with time $V_{\text{ext}}(t)=0.0035 t$:
 \Data\FigS1\FigS1E.txt

CV1-CV2 distributions derived from the time stamps obtained by simulating stochastic equations E1a-c at high (\Data\FigS1\FigS1F-high-noise.txt) and low (\Data\FigS1\FigS1F-low-noise.txt) noise intensities. See Table ST1 for detail.

CV1-CV2 distributions derived from the time stamps obtained by simulating stochastic equations (Eqs. 1a-c) with only the bath noise (\Data\FigS1\FigS1G-bath.txt) or only the circuit noise (\Data\FigS1\FigS1G-circuit-noise.txt). See Table ST1 for detail.

----- **Fig S2 data** -----

Simulated and measured spiking response. Time is measured in seconds for measurements and in units of 'RC' time (τ) for simulations. Simulations are done using stochastic equations (Eqs. 1a-c); see Table ST1 and Section 2 of SI for detail.

Time dependence of measured voltage drop (across the transneuron) given in V [second column] and normalised by V_{ext} [third column] at $V_{\text{ext}}=1.3$, $R_{\text{ext}}=65$ kOhm, $C=50$ nF:
 \Data\FigS2\FigS2A.txt.

Time dependence of simulated transneuron conductance in units of maximum conductance:
 \Data\FigS2\FigS2B.txt

Time dependence of simulated transneuron conductance in units of maximum conductance at higher bath temperature than in Fig. S2B:
 \Data\FigS2\FigS2C.txt

----- **Fig S3 data** -----

Time stamps for measured cortical PM neurons and simulated transneurons using stochastic equations (Eqs. 1a-c). Simulation details are given in the data files, Table ST1, and Section 2 of SI.

Measured time stamps of a PM neuron, with time is given in seconds (see the data file for further detail):
 \Data\FigS3\FigS3A.txt

Simulated time stamps of a transneuron with time in units of τ . The time stamps given in \Data\FigS3\FigS3B.txt correspond to the raw data recorded in \Data\Fig2\FigS2B.txt.

Simulated time stamps of a transneuron with time in units of τ at higher bath temperature than in Fig S2B. These time stamps given in \Data\FigS3\FigS3C.txt correspond to the raw data in \Data\Fig2\FigS2C.txt.

----- **Fig S4 data** -----

Simulated distributions of ISI/ τ at different bath temperatures $T_0=0$ (\Data\FigS4\FigS4-black.txt), 0.007 (\Data\FigS4\FigS4-magenta.txt), 0.02 (\Data\FigS4\FigS4-brown.txt), 0.04 (\Data\FigS4\FigS4-green.txt), 0.1 (\Data\FigS4\FigS4-orange.txt), 0.2 (\Data\FigS4\FigS4-violet.txt). The data were obtained using Eqs. 1a-c; also see Table ST1 and Section 2 of SI.

----- **Fig S5 data** -----

CV1-CV2 by stimulus contrast obtained by measuring cortical activity in area MT (time stamps) activated by a Gabor stimulus at different luminance contrasts.

Distribution data of CV1-CV2 are given for two high contrasts in \Data\FigS5\FigS5-high-contrast.txt and for two low contrasts in \Data\FigS5\FigS5-low-contrast.txt. See the caption of Fig. S5 for detail.

----- **Fig S6 data** -----

Measured data are the same as for Fig. 5, see data files in \Data\Fig5

----- **Fig S7 data** -----

Simulated distributions of inter-spike intervals at different periods of AC voltage at the fixed DC voltage corresponding to the interval at which the self-sustained heating-cooling cycle is realised. AC oscillation period and inter-spike intervals are normalised by τ . For further detail, see the caption of Fig. S7 and Table ST1. Data are available in \Data\FigS7\FigS7A.txt.

Simulated rates (spikes per one thirtieth of τ) as a function of AC voltage normalised by the DC voltage, which is the same for all simulations, and AC oscillation period t_p in units of τ . See the caption of Fig. S7 and Table ST1 for detail. Data are available in \Data\FigS7\FigS7B.txt.

----- **Fig S8 data** -----

ISI sequence in milliseconds measured in two neurons in MT monkey cortex (which behave differently), used to construct the violin plots: \Data\FigS8\FigS8A_left.txt (for one neuron) and \Data\FigS8\FigS8A_right.txt (for another neuron)

ISI sequence in seconds measured in the transneuron at two DC voltages: one voltage brings the transneuron into the region of noise-induced oscillations (\Data\FigS8\FigS8B_right.txt) and the other into the region of self-sustained spiking (\Data\FigS8\FigS8B_left.txt)

ISI sequence (in units $\tau/30$) simulated for the transneuron at two DC voltages: one voltage brings the transneuron into the region of noise-induced oscillations (\Data\FigS8\FigS8C_right.txt) and the other into the region of self-sustained spiking (\Data\FigS8\FigS8C_left.txt)

----- **Fig S9 data** -----

Voltage drop (channel B) across the neuron and the applied voltage (channel A) both measured in V versus time in seconds, for three successive 30 sec pulses separated by 35-second idle intervals, in the transneuron with $R_{\text{ext}}=65$ kOhm and $C=50$ nF: \Data\FigS9\FigS9a.txt

The same data as in \Data\FigS9\FigS9a.txt, but split into three pulses (\Data\FigS9\FigS9b-green.txt; \Data\FigS9\FigS9b-black.txt; \Data\FigS9\FigS9b-red.txt) shifted in time so the onsets of all the pulses coincide.

Raw measured spiking data (1497 raw data files) for transneurons [either current $I(t)$ or voltage drop across memristor $V(t)$] versus time used for obtaining time stamps and deriving CV1-CV2 characteristics shown in Fig. 5.

Files names:

\Data\spiking_data_for_transneurons\sample#_series#_measurement#.dat

sample# with #=1-5 refers to different transneurons used for measurements, while series# and measurement# represent the measurement ID number. For all transneurons below, the diffusive memristors were different.

Transneuron sample 1 (fabrication method 1): $R_{\text{ext}}=70$ kOhm, $C=100$ nF
 Transneuron sample 2 (fabrication method 1): $R_{\text{ext}}=70$ kOhm, $C=100$ nF
 Transneuron sample 3 (fabrication method 2): $R_{\text{ext}}=65$ kOhm, $C=1$ nF
 Transneuron sample 4 (fabrication method 2): $R_{\text{ext}}=70$ kOhm, $C=1$ nF
 Transneuron sample 5 (fabrication method 2): $R_{\text{ext}}=65$ kOhm, $C=1$ nF

Examples of raw time stamps measured in biological neurons

The folder Data\three_cortical_areas-data_samples_for_biological_neurons\ contains 31 files:

- Ten files with samples of time stamps for each area: MT, PRR, and PM (30 files in total).
- The file *read_and_plot_example.m* contains MATLAB code that illustrates how to read the data and plot spike trains. The code is set up to read one file of each type (MT, PRR, and PM) and plot its spike train, displaying one spike train per cortical area.